

# Modelling ocean melt of ice mélange at Greenland's marine-terminating glaciers

Lokesh Jain[1], Donald A. Slater[1], and Peter Nienow[1]

[1]School of GeoSciences, University of Edinburgh, Edinburgh, UK

**Correspondence:** Lokesh Jain (lokesh.jain@ed.ac.uk)

**Abstract.** Many of Greenland's marine-terminating glaciers have retreated and accelerated in recent decades, contributing significantly to sea level rise. Increased submarine melting of calving fronts is often cited as the dominant driver of this retreat. However, the presence of ice mélange and its associated buttressing force on a glacier terminus is also thought to significantly impact glacier advance and retreat. The buttressing force depends on the mélange thickness, and thickness will be modulated

by ocean melt rate, but our understanding of the melting of ice mélange by the ocean remains limited, and it is not yet known how these melt rates vary across a range of glacial and environmental conditions. Here, we perform high-resolution numerical simulations using MITgcm to model the circulation of ocean waters through an ice mélange close to marine-terminating glaciers and estimate the resultant melt. We explore the sensitivity of mélange melt rate to environmental conditions, finding that melt rate increases sublinearly with subglacial discharge and approximately linearly with ocean temperature. In this sense,

mélange melt rate appears to respond to environmental forcing in a similar manner to submarine melting of the calving front, and can be parameterised as such. This work is a step towards both a better understanding of ice mélange dynamics and a better parameterisation of its effects on glaciers and the ocean.

## 1   Introduction

Many of Greenland's marine-terminating glaciers have retreated (Catania et al., 2018; King et al., 2020) and accelerated (Howat

et al., 2005; Joughin et al., 2012) in recent decades, contributing $10.8 \pm 0.9 \, \mathrm{mm}$ to sea level rise between 1992 and 2018 (The IMBIE Team, 2020). Understanding the interaction between the ice sheet and the ocean is key to explaining this behaviour. An increase in ocean temperatures due to a heightened influx of Atlantic water (Wood et al., 2021) is often cited as the dominant driver of glacier retreat, as warmer oceans lead to enhanced glacier submarine melting (Straneo and Heimbach, 2013) as well as increased iceberg calving (O'Leary and Christoffersen, 2013). However, recent work has shown that atmospheric warming

can be just as significant an influence on submarine melting (Slater and Straneo, 2022): surface meltwater injected at depth into the ocean as subglacial discharge drives buoyant plumes which increase the turbulent transfer of heat from the ocean to the ice (Holland and Jenkins, 1999; Jenkins, 2011; Carroll et al., 2015). Whilst both oceanic and atmospheric forcing are important factors underpinning the behaviour of Greenland's glaciers, there is a third component which can have a substantial effect on the cycles of glacial advance and retreat: the presence or absence of ice mélange.





Ice mélange is a mixture of icebergs, bergy bits and sea ice which can be found in front of some tidewater glaciers (Straneo et al., 2016) and forms when the combination of ocean currents and surface winds are unable to remove icebergs from a fjord sufficiently quickly (Burton et al., 2018). As a result, a fjord with constrictions and a glacier with a high calving rate are more likely to have ice mélange. For some Greenlandic glaciers, ice mélange forms sporadically when temperatures are low enough for thick sea ice to form, locking the icebergs in place (Burton et al., 2018), such as in the Uummannaq district of

West Greenland (Howat et al., 2010). However, larger glaciers may have ice mélange which persists year round: this is the case for glaciers such as Sermeq Kujalleq (Jakobshavn Isbræ) (Amundson et al., 2010) and Helheim (Wehrlé et al., 2023). Nevertheless, the colder winter temperatures and sea ice growth which comes with it may serve to strengthen this permanent mélange (Robel, 2017). Indeed, as a granular material, the rheology of ice mélange changes seasonally due to the thickening and thinning of sea ice with changing temperatures (Amundson et al., 2010).

The presence of an ice mélange can have several impacts on a glacier and its fjord. Firstly, if coherent enough, it may exert a buttressing force onto the glacier terminus, typically transmitting stresses from the fjord walls backwards (Burton et al., 2018). This buttressing force increases with mélange thickness – a thicker mélange transmits more stress and is more resistant to stress-induced breaking (Robel, 2017). If the average force per unit width transmitted to the glacier terminus is sufficiently high (typically around $10^7 \, \mathrm{N \, m^{-1}}$), this buttressing effect can cause calving to cease (Robel, 2017). If an ice mélange subsequently

melts, it will thin and weaken, reducing the backstress provided by the mélange onto the glacier. This means calving is more likely to occur, initiating glacier retreat (Bevan et al., 2019). Indeed, the numerical modelling of Krug et al. (2015) suggests that ice mélange has a greater impact on the seasonal cycles of a glacier front than submarine melting, and observations of Sermeq Kujalleq by Joughin et al. (2020) suggest that the primary link between ice flow speed and ocean forcing is via mélange rigidity. Amundson et al. (2024) use a model of stress in proglacial mélange to show that the buttressing force depends on the

basal mélange melt rate through an inverse power law relationship with an exponent of about $-3$, demonstrating the potential for small changes in ocean melt rate to have a large impact on the buttressing force.

Secondly, the melting of a mélange inputs a substantial flux of freshwater into the fjord. This meltwater flux can exceed that provided by subglacial discharge and submarine melting of the glacier terminus (Enderlin et al., 2016), and in one modelling study, iceberg melt was the largest annual freshwater source in the Helheim Glacier – Sermilik Fjord system (Moon et al., 2018).

The cooling and freshening triggered by this submarine iceberg melting is generally localised in the upper water column, with the polar water located at intermediate depths typically warming due to iceberg melt-induced upwelling of warm Atlantic water (Davison et al., 2022). Moreover, the mélange meltwater flux may even deepen the neutral buoyancy depth of the subglacial plume (Kajanto et al., 2023). This not only lowers the depth at which the glacially-modified waters are exported down-fjord, but also suppresses plume-driven glacier melting in the upper layer of the water column, thereby potentially promoting undercutting

of the calving front.

The freshwater released by the melting of an ice mélange can also drive a fjord-wide circulation which increases the net up-fjord heat flux (Davison et al., 2020). Indeed, the analytical model of icebergs melting in a rectangular fjord developed by Hughes (2024) suggests that a given iceberg-induced meltwater flux can induce a fjord-wide exchange flow which is up to 50 times larger than the meltwater flux itself. The presence of ice mélange also inhibits the flow of fjord water, slowing the



near-surface current and causing the fastest current speeds to be found at or below the drafts of the deepest icebergs (Hughes, 2022).

Observational estimates of the ocean melt rate of ice mélange are limited but the magnitudes obtained support the idea of mélange melt being a significant term determining mélange thickness. Enderlin et al. (2016) used remote sensing methods at Ilulissat Isfjord and Sermilik Fjord to estimate a melt rate of approximately $0.1 - 0.8 \,\mathrm{m\,d^{-1}}$. Other observations of iceberg
meltwater flux in Sermilik Fjord have demonstrated a strong seasonal signal peaking across August and September (Moyer et al., 2019b), whilst observations of the seasonal ice tongue at the glacier Kangiata Nunaata Sermia suggest an estimated melt rate of approximately $1 \,\mathrm{m\,d^{-1}}$ (Moyer et al., 2017, 2019a).

With regards to modelling the ocean melt of ice mélange, studies have typically focused on how meltwater flux from icebergs influence large-scale fjord circulation and dynamics (Davison et al., 2020, 2022; Kajanto et al., 2023). Such studies examining
the fjord-scale impact of iceberg melting typically use the 'IceBerg' package (Davison et al., 2020) as implemented in the ocean modelling software MITgcm to approximate iceberg melting. By assigning an iceberg concentration to each grid box and estimating the resultant melting by using the velocity-dependent three equation formulation (Jenkins, 2011), the IceBerg package freshens, cools and decelerates the ocean water in that grid box accordingly. However, this coarse approximation of the iceberg-melting process, necessary to run fjord-scale simulations (with a typical horizontal resolution of $300 - 500 \,\mathrm{m}$) in a
reasonable computational time, is unable to resolve exactly how water flows between individual icebergs. One of the key tenets of the three equation formulation is that the melting of ice by the ocean depends on the velocity of the ocean; and since these studies do not resolve the velocity of the ocean around icebergs in high resolution, they are unable to precisely analyse the flow and thermodynamics related to ocean-induced melting of ice mélange, resulting in uncertainty in their melt rate estimates.

Hughes (2022) modelled ocean flow through an ice mélange at $10 \,\mathrm{m}$ horizontal resolution, and was therefore able to resolve
the ocean velocity in greater detail. He found that the presence of cuboid icebergs slows the flow in some places, particularly near the surface, but accelerates it in others. However, this study only examined the effect of the physical presence of an ice mélange on the flow through a fjord without including the effect of any melting processes. Since the melt rate of ice depends on the water velocity (Holland and Jenkins, 1999), the variable impact which the presence of icebergs has on the speed of the flow means that the net effect an ice mélange will have on melt rates is not clear. Understanding in detail what factors are
key in determining mélange melt rates is crucial to justifying the assumptions made in the IceBerg package and updating the parameterisations used in fjord-scale representations of iceberg melting. Hughes (2024) modelled the melting of icebergs in an ice mélange, including thermodynamic processes, and found melt rates of $0.3 - 0.4 \,\mathrm{m\,d^{-1}}$. This melt induces a flow of order $5 \,\mathrm{cm\,d^{-1}}$, and the fastest flows occur in localised hotspots near the surface where water is squeezed through the gaps between icebergs. Hughes (2024) mainly focused on the fjord circulation induced by melting icebergs but a systematic analysis of the
factors affecting the melt rate of an ice mélange has not yet been done.

High resolution mélange melting simulations are therefore required to understand the fundamental processes behind the melting of ice mélange. In particular, it is important to gain a deeper understanding of how mélange melting varies under different glacial and environmental conditions – how sensitive are mélange melt rates to ocean temperatures, subglacial discharge flux, or the geometry of the mélange itself? Here, we model the melting of an artificially-created ice mélange using MITgcm





to examine the sensitivity of the ocean melt rate of ice mélange (hereafter referred to as the mélange melt rate) to changes in glacial and environmental conditions. This paper proceeds as follows. Section 2 describes the methods, including the model setup, boundary conditions, and simulation details. Section 3 presents the results of our simulations. Section 4 discusses the implications of our findings, and section 5 concludes the study and considers future research directions.

## 2  Methods

### 2.1  Model setup

We model the melting of an ice mélange using a model domain of $15 \text{ km} \times 5 \text{ km} \times 500 \text{ m}$ ($600 \times 250 \times 50$ grid points) using the three-dimensional, non-hydrostatic ocean model MITgcm (Marshall et al., 1997a, b) at 20 m horizontal and 10 m vertical resolution within the mélange (Figure 1a). The ice mélange itself has dimensions of $10 \text{ km} \times 5 \text{ km}$. The size of the domain and of the ice mélange has been chosen based on the Helheim Glacier – Sermilik Fjord system but is intended to be representative of large Greenlandic glacier-fjord systems. Working from west to east, the first layer of grid cells $x_1$ represents the calving front of a glacier. This layer is, therefore, made up entirely of ice cells, apart from a central $200 \text{ m} \times 30 \text{ m}$ channel where subglacial discharge emerges. The second layer $x_2$ is open water; the layers $x_j$ for $3 \leq j \leq 502$ are made up of an artificially-created ice mélange consisting of cuboid icebergs of a range of different shapes and sizes (see section 2.3 for more details). The eastern-most 5 km of the domain, layers $x_k$ for $503 \leq k \leq 600$, is a restoring region, consisting of open water and a sponge layer to allow the fluid flow to relax to the boundary conditions before exiting the domain. The resolution of this restoring region gradually decreases away from the mélange. The Coriolis parameter is set for 70°N ($f = 1.37 \times 10^{-4}\text{s}^{-1}$).

### 2.2  Initial and boundary conditions

On the western side of the domain, fresh subglacial discharge emerges from beneath the glacier at a flux $Q_0$ at the pressure melting point (approximately $-0.4$°C at a depth of 500 m; Figure 1a).

The eastern boundary temperature $T_0$ and salinity $S_0$ profiles (Figure 1b) are set by CTD profiles collected from near the end of the mélange in Sermilik Fjord, southeast Greenland (Straneo et al., 2011). These profiles demonstrate the presence of a warm, deep and salty Atlantic water layer in the bottom 400 m of the fjord, with a cooler and fresher surface layer above this in the upper 100 m. In order to test the sensitivity of mélange melt to fjord temperature, we also generate anomalous positive and negative temperature profiles $T_{\pm\gamma}$ via

$$T_{\pm\gamma}(z) = T_0(z) \pm \frac{\gamma}{2}\left[\tanh\left(\frac{z-100}{30}\right) + 1\right] \tag{1}$$

where $\gamma = 1.0$°C or $2.0$°C is the magnitude of the desired temperature anomaly; $z$ is the (positive) depth below the surface; 100 m is the target depth at which the anomalous temperature profile should start diverging from the baseline temperature profile; and 30 m is the approximate height of the layer over which the temperature profile changes from the baseline profile to the anomalous profile (Figure 1b). We set a depth at which the temperature profile should start diverging as we do not wish





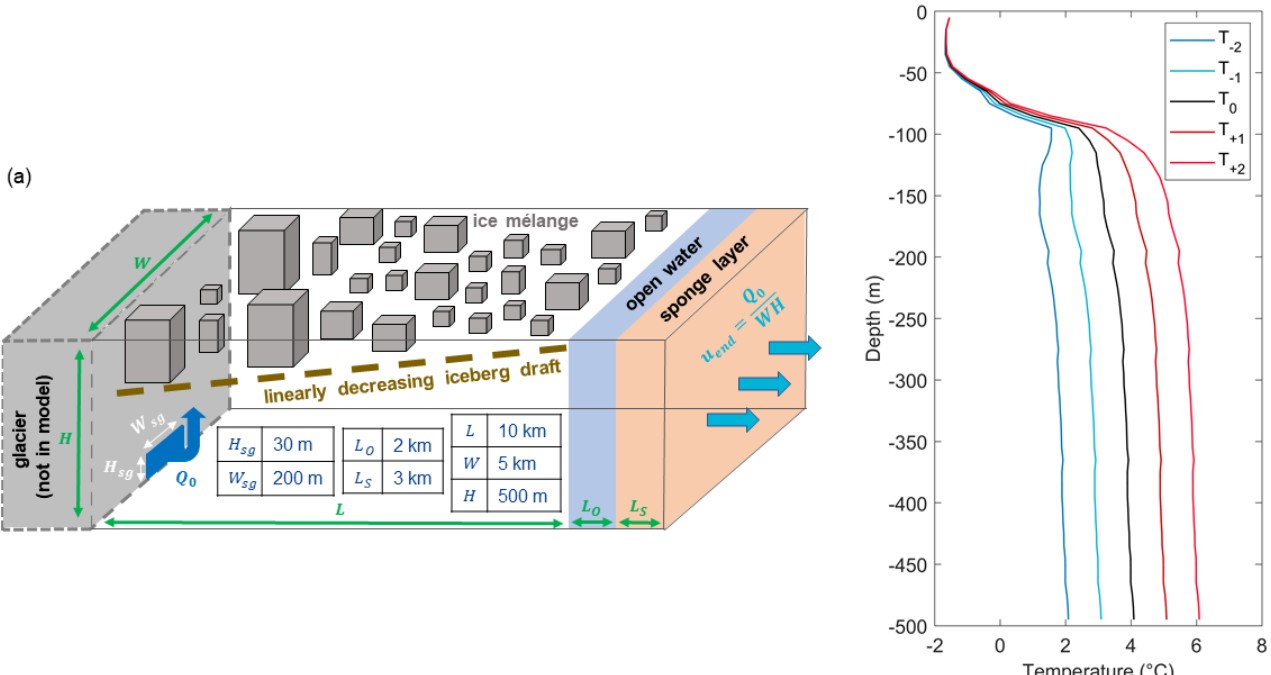

**Figure 1.** (a) A schematic of the model set up. The fjord is 15 km × 5 km × 500 m. The ice mélange (10 km× 5 km) has a linearly decreasing draft away from the glacier terminus and is represented by cuboid icebergs using the MITgcm packages ShelfIce and IceFront. Subglacial discharge $Q_0$ emerges from beneath the calving front from a channel of size $H_{sg} \times W_{sg}$. A sponge layer is implemented on the eastern side of the domain to gradually relax the fluid flow to initial conditions before exiting the domain. (b) The various temperature profiles used in this study. A standard temperature profile $T_0$ is taken from observations from Sermilik Fjord (Straneo et al., 2011); the other profiles are generated according to equation 1.

to change the initial temperature of the surface waters between simulations. The reason for this is that we expect water near the surface will be very strongly affected by the melting of the ice mélange, whereas water at depth is upwelled by the plume. We therefore vary the temperature of these deeper waters which are the source of the heat which drives the melting of the ice mélange.

     The initial conditions of a particular simulation are set to be the same as the boundary conditions. However, because the

simulations are run until they have reached approximate steady state (section 2.6), we note that the model is insensitive to this initialisation; the simulations are run for long enough that the model effectively forgets the initial conditions. The final state which the model evolves to is dependent only on the boundary conditions and the fixed mélange geometry.



## 2.3 Making a mélange

In order to investigate systematically how mélange geometry affects the melting of mélange, we sought a way to generate
an artificial mélange whose properties we could then vary between simulations. The icebergs which constituted this artificial
mélange had to have certain realistic characteristics. Firstly, the artificial icebergs had to replicate observed iceberg size power
law distributions $N \propto A^{-\alpha}$, where $N$ is the number of icebergs of horizontal surface area $A$ (Sulak et al., 2017; Shiggins et al.,
2023). Secondly, the artificial icebergs had to have realistic aspect ratios such that the volume $V$ of an iceberg is related to its
area $A$ by $V = aA^b$, with the 95% confidence interval for $a$ and $b$ being $3.4 < a < 8.6$ and $1.26 < b < 1.33$ (Sulak et al., 2017).
Thirdly, the artificial icebergs had to have a mean draft which varied down-fjord in agreement with observations of mélange
thickness (Enderlin et al., 2016; Meng et al., 2024).

In order to respect all of these constraints, we randomly sample icebergs from the following power law distribution for the
number of icebergs $N$ of draft $H$ (see appendix A for more details):

$$N(H) = \left( \frac{-\beta + 1}{H_{\max}^{-\beta+1} - H_{\min}^{-\beta+1}} \right) H^{-\beta} \tag{2}$$

where $H_{\min}$ and $H_{\max}$ are the smallest and largest iceberg draft in the distribution, and $\beta$ varies with distance away from the
glacier terminus. Equation 2 has been derived assuming the aforementioned iceberg size power law distribution and realistic
aspect ratios, and the value of $\beta$ is determined based on observations of mélange thickness.

We randomly sample icebergs from the distribution using inverse transform sampling (appendix B) until the iceberg areal
fraction $\lambda$ (i.e. the percentage areal coverage of the fjord surface covered by icebergs) reaches 0.6. There are very few estimates
of $\lambda$ within a mélange: Foga et al. (2014) use satellite remote sensing techniques at Helheim Glacier to determine an iceberg-
to-sea ice ratio of between 0.0 and 0.35; assuming a mélange is made up entirely of icebergs and sea ice, this corresponds to
$0.0 \leq \lambda \leq 0.54$. Since there is much uncertainty and variability in this value, we set $\lambda = 0.6$ in all model runs. These artificial
icebergs are then placed into the domain within MITgcm via the ShelfIce package (Losch, 2008). The icebergs both inhibit the
flow of water and generate freshwater via melting (section 2.4).

The standard mélange profile used in the simulations (Figure 2a) has a mean iceberg draft of $100\,\mathrm{m}$ at the calving front and
$50\,\mathrm{m}$ at a distance of $10\,\mathrm{km}$ down-fjord, motivated by observations of the ice mélange in Sermilik fjord (Enderlin et al., 2016).
There is much variability in the sizes of icebergs everywhere in the domain, as is to be expected – large icebergs occasionally
travel a long way down-fjord, and small icebergs are also found very close to the glacier terminus. Whilst the icebergs generated
in this segment follow the underlying target probability distribution quite closely (Figure 2b), the icebergs actually placed in
the mélange diverge from this slightly. If the domain were an infinite size then this would not be a problem; however, trying
to squeeze all of the generated icebergs into a finite space whilst also ensuring $\lambda = 0.6$ means that some deviation is likely to
occur.

In addition to the standard mélange profile, we also run simulations with a thick mélange profile and a thin mélange profile.
We do this to investigate the effect that the thickness of a mélange has on the simulated melt rate of the icebergs in the mélange.





The thickness of the artificial mélange is adjusted by varying the parameter $\beta$ accordingly in equation 2 (appendix A). The thick (thin) profile has a mean draft of $200\,\mathrm{m}$ ($50\,\mathrm{m}$) at the calving front and $100\,\mathrm{m}$ ($25\,\mathrm{m}$) at a distance of $10\,\mathrm{km}$ down-fjord. These values are chosen based on the range of observed mélange thicknesses at both Helheim Glacier and Sermeq Kujalleq (Enderlin et al., 2016; Meng et al., 2024). The mean thickness profile for all mélange configurations is very close to the target mean draft (Figure 2c).

Note that the mélange we generate here does not include sea ice. This is because we do not expect sea ice to contribute significantly to the mélange meltwater flux – as a thin layer near the surface, the water surrounding sea ice is predominantly cool and stagnant, and so we do not expect sea ice to melt significantly compared to the icebergs in a mélange.

## 2.4  Representation of melting

We represent iceberg melting using the typical three-equation melt parameterisation (Holland and Jenkins, 1999; Jenkins,
2011) which ensures that, at the ice-ocean interface: (a) heat flux through the ice-ocean boundary layer is balanced, (b) salt flux through the ice-ocean boundary layer is balanced, and (c) the temperature of the ice-ocean boundary layer is at the pressure melting point. However, this parameterisation relies on a number of constants whose values are plagued with uncertainty (Zhao et al., 2024).

One such constant is the drag coefficient, $C_d$. In the three-equation melt parameterisation, the heat flux to the ice is strongly
controlled by the drag coefficient, with the melt rate of the ice $m \propto \sqrt{C_d}$, and so an accurate representation of drag in a model is crucial in order to simulate melt rates realistically. A value of $C_d = 2.5 \times 10^{-3}$ is typically used (Jenkins et al., 2010). However, some studies have found that using this value of the drag coefficient underestimates melt rates at glacier faces (Sutherland et al., 2019; Jackson et al., 2020). Moreover, as Zhao et al. (2024) argue, using a single value for the drag coefficient does not distinguish between a range of possible flow scenarios at the ice-ocean interface along the face of an
iceberg. The heat flux to the ice will likely depend on whether the interface is horizontal or vertical. The heat flux to the ice will also depend on whether the fluid flow is primarily driven by plume-driven convection, melt-driven convection, or a general fjord-wide horizontal flow and overturning circulation (Zhao et al., 2024). It is therefore important to adjust the value of $C_d$ in the melt rate parameterisation according to the fluid flow in a particular grid cell in order to accurately represent iceberg melting in MITgcm.

The current three-equation melt parameterisation could be improved by incorporating the interface slope and driver of the fluid flow directly into the determination of the ice melt rate. However, as a temporary solution, we suggest that the modelled velocity itself can be used as an effective mask that identifies different regimes of melting. In particular, modelled regions undergiong melt-driven convection have ice-adjacent velocities less than $0.04\,\mathrm{m\,s^{-1}}$, and the only place in the simulations where fluid velocities are greater than $1\,\mathrm{m\,s^{-1}}$ is in the plume itself. We therefore assign values of the drag coefficient based
on Zhao et al. (2024) and the modelled ice-adjacent velocity in a particular grid cell:

1. Melt-driven convection: $|u| \leq 0.04\,\mathrm{m\,s^{-1}}$, $C_d = 0.15$

2. Horizontal flow: $0.04\,\mathrm{ms^{-1}} < |u| \leq 1.00\,\mathrm{m\,s^{-1}}$, $C_d = 0.0025$



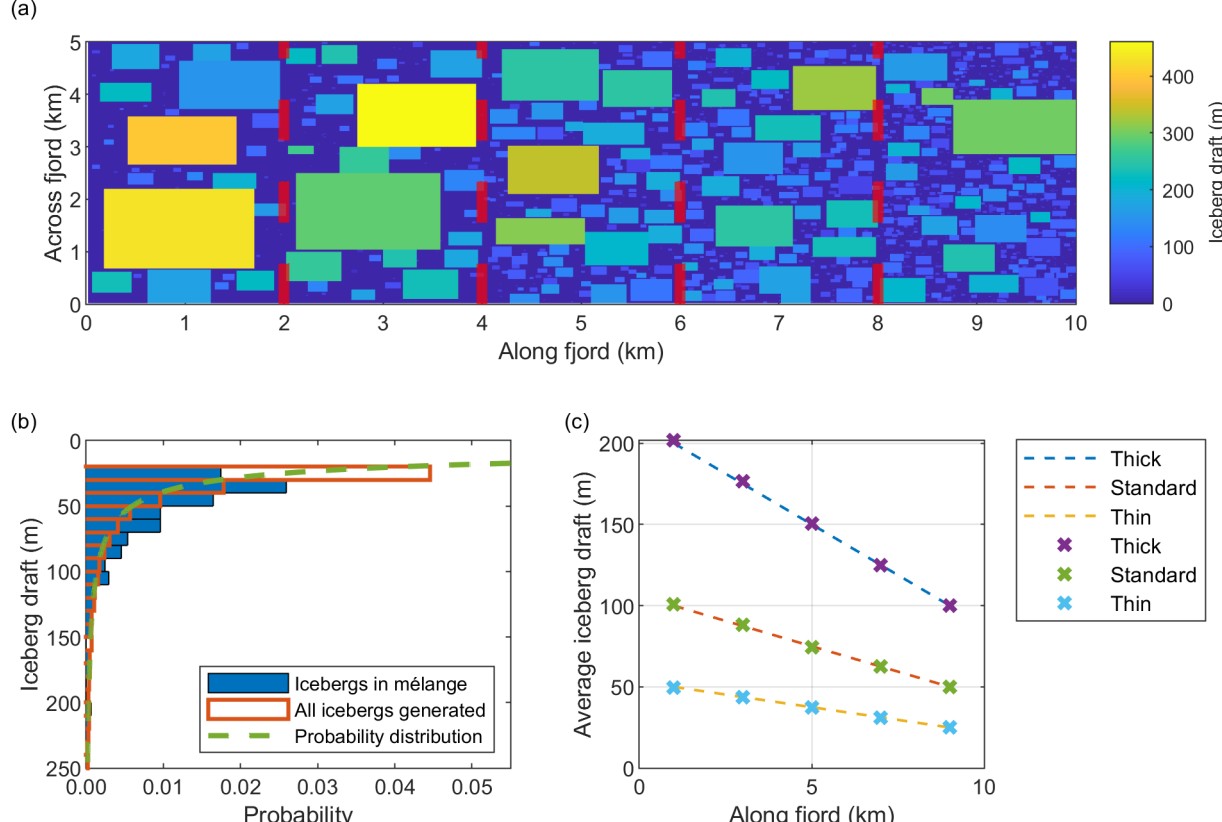

**Figure 2.** (a) A plan view of the standard mélange profile used in the simulations. The mélange is split into five segments (delineated by the dashed red lines) each with a length along fjord of 2 km. Each segment has a distinct mean iceberg draft; this mean average draft decreases down fjord to represent the thinning of a mélange. (b) The distribution of icebergs in the final segment with a mean iceberg draft of 50 m. In orange are all the icebergs generated; in blue are all the icebergs actually placed in the mélange. Not all generated icebergs are placed because icebergs stop being placed once the iceberg areal fraction $\lambda = 0.6$. In green is the underlying probability distribution from which the icebergs are generated. (c) The mélange profiles target mean draft (dashed line) and actual mean draft (crosses) for the thick, standard and thin profiles.

3. Plume-driven convection: $|u| > 1.00 \, \mathrm{m\,s^{-1}}$, $C_d = 0.015$

where $|u|$ is the velocity magnitude accounting for both horizontal and vertical components. We impose a minimum back-
ground velocity of $0.04 \, \mathrm{m\,s^{-1}}$ at each lateral iceberg face to represent unresolved melt-driven convection (Cowton et al., 2015).
In practice, what this means is that we assume melt-driven convection occurs everywhere; if the velocity of fluid at the ice-
ocean interface is calculated to be above $0.04 \, \mathrm{m\,s^{-1}}$, then we enter the horizontal flow regime; and if the velocity of the fluid
is above $1 \, \mathrm{m\,s^{-1}}$, then we enter the plume-driven convection regime.



Whilst this method has the disadvantage of not being very elegant (an all-encompassing melt rate parameterisation would be more rigorous), it has the advantage of being consistent both with Zhao et al. (2024) and also with other literature suggesting that the standard value of $C_d = 2.5 \times 10^{-3}$ underestimates the melt rate at the ice-ocean interface (Sutherland et al., 2019; Jackson et al., 2020).

We represent melting in MITgcm using the ShelfIce (Losch, 2008) and IceFront (Xu et al., 2012) packages for basal and lateral melting respectively. We set the turbulent transfer coefficients of heat and salinity to $\Gamma_T = 2.2 \times 10^{-2}$ and $\Gamma_S = 6.2 \times 10^{-4}$ respectively (Jenkins, 2011).

## 2.5 Introduction of key analysed quantities

We analyse the simulations in terms of the modelled circulation, the modelled temperature, the mean mélange melt rate and the total mélange meltwater flux. The latter two deserve a specific definition since their calculation must take account of the geometry of the mélange. For each ocean grid cell $i$, we calculate the total meltwater flux $\phi_i$ into that grid cell as $\phi_i = m_{i,s}A_{i,s} + m_{i,b}A_{i,b}$ where $m_{i,s}$ [$m_{i,b}$] and $A_{i,s}$ [$A_{i,b}$] are the melt rate and area of ice on the sides [bottom] of icebergs available for melting in grid cell $i$. We then define the mean melt rate $m_i$ in grid cell $i$ as $m_i = \phi_i/A_i$ where $A_i = A_{i,s} + A_{i,b}$ is the total area of ice available for melting in a given grid cell.

The total mélange meltwater flux is then defined as $\Phi = \sum_i \phi_i$ and the mean mélange melt rate is defined as $m = \sum_i m_i/N$ where $N$ is the number of ocean grid cells in contact with ice. Since we wish to focus on melting of the ice mélange, melting of the calving front is not included in our analysed quantities.

## 2.6 Simulations run

We vary the subglacial discharge flux $Q_0$, the temperature profile of the fjord and the mélange thickness profile between simulations and we analyse the subsequent impact on mélange melt rates. We test values of $Q_0$ of $10, 30, 100, 300, 600$ and $1000 \ \mathrm{m^3 s^{-1}}$ as these are similar to those modelled and observed at a large Greenlandic glacier such as Helheim throughout the year (Mankoff et al., 2020; Karlsson et al., 2023). We also include a $Q_0 = 0 \ \mathrm{m^3 s^{-1}}$ simulation with the standard mélange and temperature profiles to isolate the circulation and iceberg melting that is driven by the presence of ice mélange alone. As discussed in section 2.2, the temperature profiles are generated via equation 1 to give a range of thermal forcing scenarios. We use a standard, thick and thin mélange profile.

All simulations run as part of this study are shown in Table C1. Rather than simulate every possible permutation of subglacial discharge, temperature and mélange thickness, we instead choose only to vary the temperature profiles and mélange thickness for subglacial discharge values of $Q_0 = 10, 300$ and $1000 \ \mathrm{m^3 s^{-1}}$.

The simulations are run until they have reached approximate steady state, which we define as when both the (a) water temperature and salinity and (b) mélange melt rate and meltwater flux do not vary significantly from one timestep to another. Most simulations reach steady state after approximately six days of simulation time. The results throughout section 3 are averages taken whilst the model is in steady state, typically over the last 17 hours of a model run.





## 3 Results

Results are presented as follows: in section 3.1, we show the qualitative features of a representative simulation (specifically, the standard mélange thickness, $Q_0 = 300 \text{ m}^3\text{s}^{-1}$ and $T_0$ temperature profile simulation ms_q300_t0 as specified in Table C1). In section 3.2, we consider all simulations, covering the sensitivity of mélange melting to subglacial discharge, fjord temperature

and mélange thickness, before attempting to find a parameterisation that captures these dynamics.

### 3.1 Single simulation

#### 3.1.1 Circulation and properties

The influx of subglacial discharge and the resultant buoyant plume sets up a general overturning circulation in the middle of the fjord, with down-fjord flow in the upper 200 metres of the domain and an up-fjord flow layer beneath this (Figure 3a). This

down-fjord flow is fastest in the first couple of kilometres downstream from the glacier front. The water flow driven by the subglacial discharge is highly disrupted by the presence of an ice mélange (Figure 3b). In particular, the horizontal flows have to find open pathways through the icebergs in order to travel down-fjord, and this results in acceleration of the flow in some places as the water is squeezed between icebergs, and deceleration of the flow in other places. The specific pathways of fastest water flow are highly dependent on the specific configuration of each ice mélange.

The subglacial discharge plume upwells warmer waters from depth into the upper water column in the first 2 km of the domain (Figure 3c). These warm waters do not completely fill the region of the domain occupied by the mélange, which indicates that the mélange is a heat sink for the plume; as the plume advances down-fjord, the mélange extracts heat from it for melting until all of the heat has been lost. Note that the warm water in the first 2 km of the domain (Figure 3c) is not a transient feature whereby a pulse of warm water is gradually moving down-fjord; it is a steady state feature. Further along, the

temperature profile in the middle of the fjord is remarkably similar to the initial conditions of the model, with only pockets of warm water upwelling into the upper layers.

There is a general upwelling of water around icebergs seen as a 'halo' of red surrounding each iceberg (Figure 3d). This is due to melting on the sides of each iceberg forming little buoyant plumes of meltwater which rise up the iceberg face. As discussed in section 2.4, this melt-driven convection is not fully resolved in the model and so is parameterised by imposing a

minimum background velocity of $0.04 \text{ m s}^{-1}$ at each lateral iceberg face. Also visible is an alternating pattern of upward and downward velocities near (within ~1 km of) the subglacial discharge outlet – this is most likely a wave set up by the plume as it travels down-fjord and oscillates around its level of neutral buoyancy.

Averaging across the fjord and within the top 200 m of the water column, the velocity signal of the flow through the mélange fluctuates significantly due to the presence of icebergs (Figure 4). The large variation in iceberg size and number in any given

horizontal slice through the domain leads to much variability in ocean velocity as the flow navigates its way through the ice mélange. On top of this variability is a clear decreasing down-fjord trend in velocity (Figure 4a) and a peak in the middle of the fjord where the subglacial discharge outlet is located (Figure 4b). The peak velocity depth is located at approximately 100 m below the surface (Figure 4c). This corresponds to the depth at which the plume outflows down the fjord and is determined



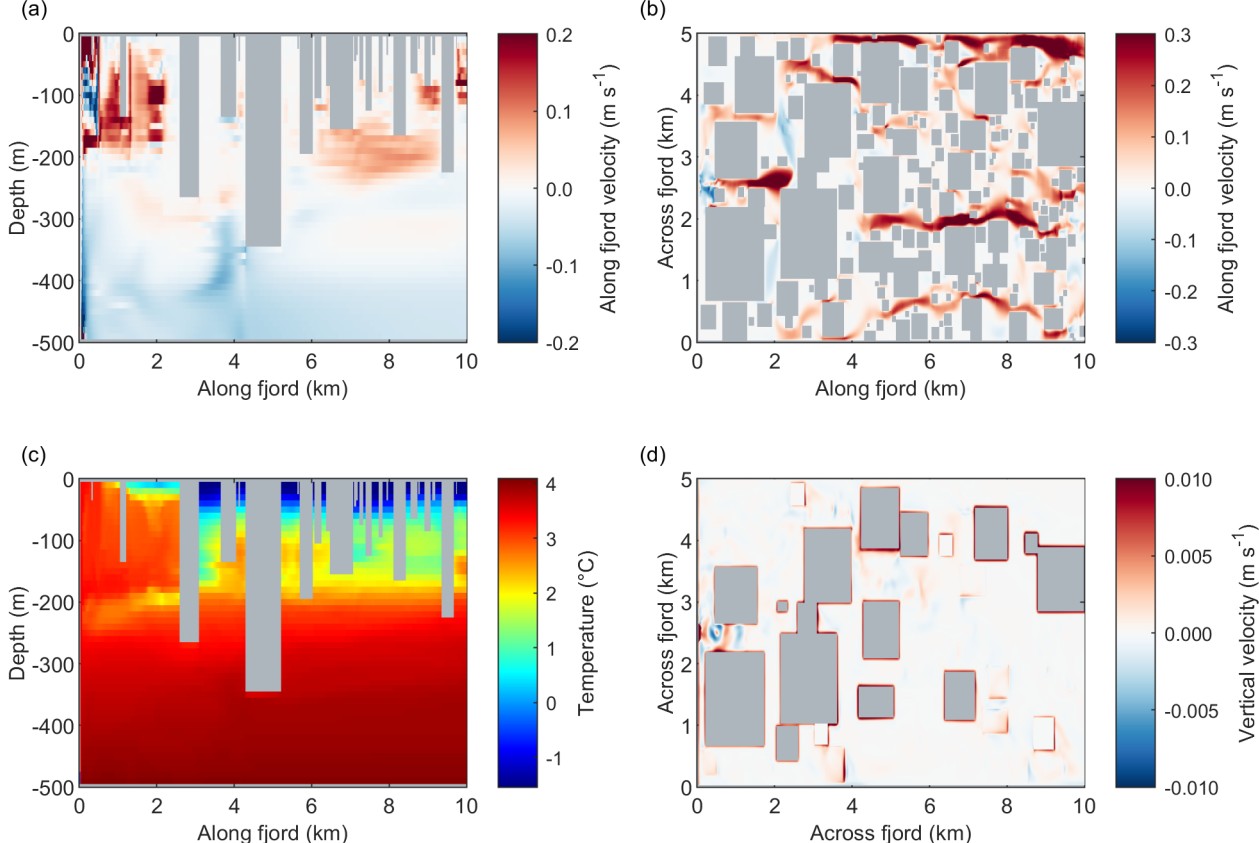

**Figure 3.** Along fjord and plan view plots of the mélange illustrating global circulation and temperature profile at the end of the model run. Icebergs within the mélange are shown as dark grey. The figures are generated by averaging over the last 17 hours of the model run, whilst the model is in steady state. (a) An along fjord plot of along fjord velocity from the middle of the fjord at y = 2.5 km. (b) A plan view plot of along fjord velocity from a depth of 100 m. (c) An along fjord plot of temperature from the middle of the fjord at y = 2.5 km. (d) A plan view plot of vertical velocity at a depth of 250 m.

by both the level of neutral buoyancy of the plume and the density of icebergs in the mélange, since the plume needs to be able
to navigate around icebergs efficiently to advance down-fjord.

The upwelling of warm water driven by the buoyant plume leads to a warming of up to $2°C$ in the upper 100 m of the fjord (Figure 4c). Temperature decreases down-fjord away from the glacier terminus (Figure 4a) due to the mélange extracting energy from the plume for melting; as the available energy decreases, so therefore does the mean water temperature. There is also cooling with respect to the initial conditions at mid-depths ($\sim$100-400 m) (Figure 4c) due to iceberg melting and mixing
of cooler surface waters downwards.





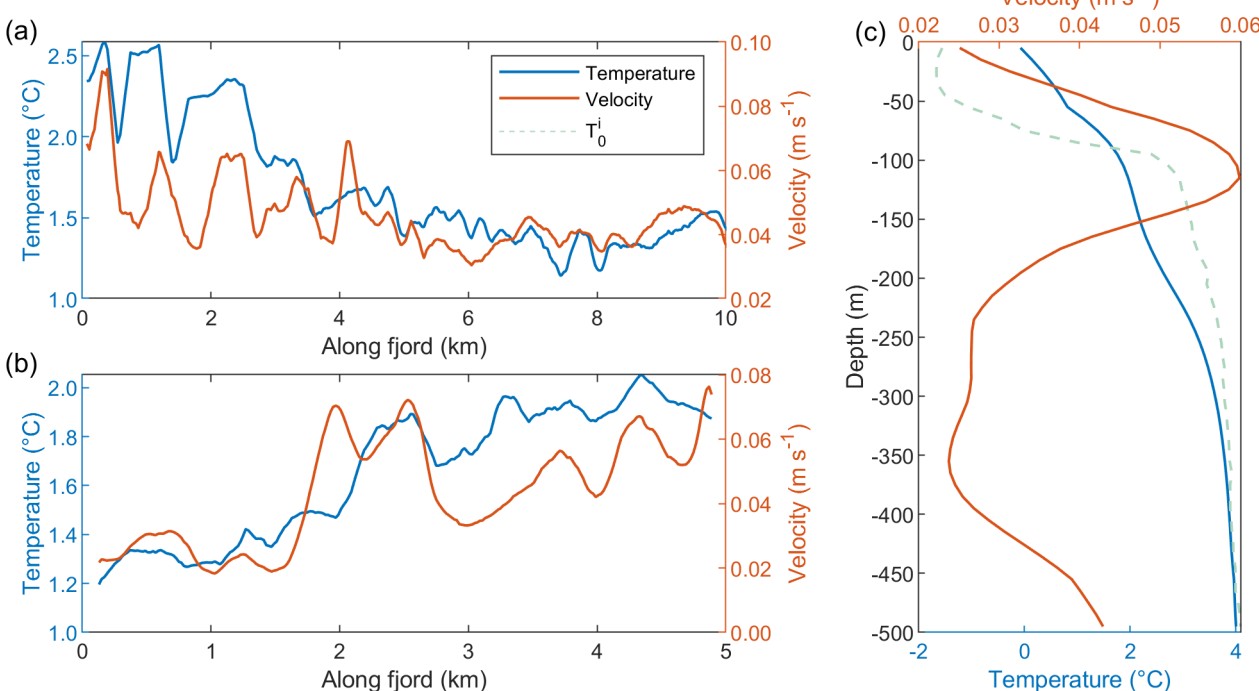

**Figure 4.** Along fjord (a), across fjord (b) and depth (c) variation of temperature and velocity magnitude throughout the mélange at the end of the model run. All variables are smoothed using a moving average with a window size of 10 grid cells, which is 200 m for the horizontal plots (a) and (b) and 100 m for the vertical plot (c). The velocity magnitude is $V = \sqrt{u^2 + v^2 + w^2}$ where $u$, $v$ and $w$ are the $x$, $y$ and $z$ components of the velocity respectively. The along fjord and across fjord plots are averaged over the upper 200 m of the fjord, which is the depth relevant to the melting of the ice mélange.

### 3.1.2 Melt rates and fluxes

The mélange melt rate decreases with distance down-fjord (Figure 5a), due partly to decreasing ocean temperature and velocity (Figure 4a), but also due to the fact that the mélange gets thinner with distance down-fjord and is therefore sitting in shallower, cooler waters. The depth at which melt rates are highest is approximately 400 m (Figure 5c) due to a combination of high

vertical velocity as a result of water upwelling from deep icebergs (Figure 3d) and warm ocean temperatures at this depth (Figure 4c). The average melt rate across the entire mélange is $m = 0.59\,\mathrm{m\,d^{-1}}$.

There is considerable fluctuation in the mélange meltwater flux due to variation in iceberg size and number throughout the domain, but the flux decreases slightly with distance down-fjord (Figure 5a). The meltwater flux is a sum of the melt rate multiplied by the surface area available for melting. Far away from the glacier terminus, low melt rates (Figure 5a) combine

with the high surface area of lots of small, shallow icebergs (Figure 2) to maintain a similar meltwater flux to the region near the glacier terminus. The depth at which the meltwater flux is highest is approximately 100 m (Figure 5c); despite melt rates being





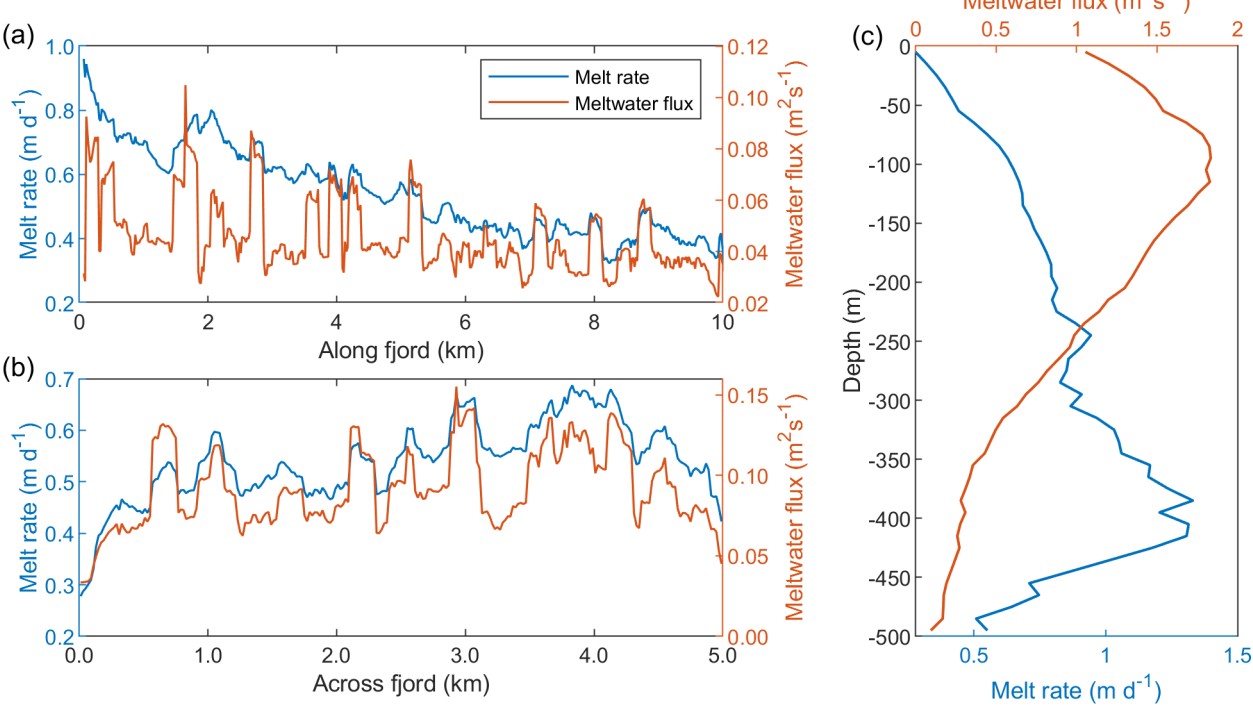

**Figure 5.** The average melt rate and total meltwater flux (a) along the fjord (b) across the fjord (c) with depth. The meltwater flux is given in units of $\mathrm{m^2s^{-1}}$ (i.e. $\mathrm{m^3s^{-1}}$ per unit distance), so that the total meltwater flux is the area under the curves.

relatively low at this depth, the large area of ice available for melting results in the high meltwater flux. The total meltwater flux is $\Phi = 510 \ \mathrm{m^3s^{-1}}$ and the total submerged surface area of ice available for melting is $A = 79 \ \mathrm{km^2}$.

## 3.2 All simulations

In sections 3.2.1–3.2.3, the influence of varying the subglacial discharge, ambient ocean temperature and mélange thickness on the steady state water temperature, water velocity, mélange melt rate and mélange meltwater flux is discussed. In section 3.2.4, the mean melt rate and total meltwater flux across all simulations is analysed, and a parameterisation for mélange melting is developed in section 3.2.5.

### 3.2.1 Sensitivity to subglacial discharge

All simulations, irrespective of the magnitude of the subglacial discharge, show cooling with respect to the initial temperature profile beneath a depth of 100 m and warming above this depth (Figure 6a), similar to the single simulation discussed in section 3.1.2 (Figure 4c). However, above 150 m the temperature profiles diverge with higher subglacial discharge resulting in





warmer ocean waters in the upper water column. This is because higher subglacial discharge generates a more vigorous plume, entraining more warm water from depth.

A higher subglacial discharge leads to a higher velocity magnitude above a depth of around 150 m (Figure 6b). A larger subglacial discharge flux generates a stronger plume, which in turns sets up a stronger fjord-wide circulation. The depth of the peak velocity is likely a reflection of the neutral buoyancy depth of the outflowing plume. This level of neutral buoyancy increases in height with increasing subglacial discharge: for $Q_0 = 10 \, \mathrm{m^3 s^{-1}}$ the level of neutral buoyancy is at a depth of 140 m, whereas for $Q_0 = 1000 \, \mathrm{m^3 s^{-1}}$ its depth is approximately 90 m.

Below a depth of 150 m, both the melt rate and the meltwater flux are insensitive to the magnitude of subglacial discharge (Figure 6c). This suggests that the melting of deeper-drafted icebergs are unaffected by changes in subglacial discharge. In the upper 150 m, larger subglacial discharge leads to both higher melt rates and meltwater fluxes, due to a combination of higher temperatures and higher velocities here. The $Q_0 = 0 \, \mathrm{m^3 s^{-1}}$ and $Q_0 = 10 \, \mathrm{m^3 s^{-1}}$ simulations are remarkably similar, showing that the overall circulation set up by a very weak plume is comparable to the circulation in the complete absence of a plume.

This demonstrates that the melting of the ice mélange alone is itself responsible for driving a weak circulation.

### 3.2.2   Sensitivity to ambient temperature

The basic shape of the temperature profile at the end of the model run is the same regardless of the temperature anomaly (Figure 7a), with warmer deep waters resulting in mélange ocean temperatures that are warmer throughout the water column. The modelled temperature differential between profiles is the same as the imposed temperature differential at depth (i.e. the

difference in temperature between the $T_0$ and $T_{+2}$ simulations is 2°C at depth) but decreases higher up the fjord. This is likely due to the shape of the initial temperature profiles, which are the same in the upper layers of the fjord and start to diverge with depth (Figure 1b).

Higher temperatures lead to slightly higher velocities throughout the entire depth of the water column (Figure 7b). This is because warmer ocean temperatures lead to more mélange melting, which in turn results in greater upwelling on the sides of

icebergs in the mélange. This increase in upwelling (which reflects an increase in the rate of melt-driven convection) leads to a more vigorous fjord circulation. A change in the initial temperature of $\pm 2$°C only leads to a change in the velocity magnitude of $\Delta V < 1 \, \mathrm{cm \, s^{-1}}$.

A higher initial temperature profile leads to a higher melt rate and meltwater flux at all depths since there is more heat available for melting the mélange (Figure 7c). The difference in the meltwater flux between simulations is smallest at depth

(since there are very few icebergs melting at this depth so there is little area available for melting) and grows with increasing height (since there are more icebergs and so more area is available for melting).

### 3.2.3   Sensitivity to mélange thickness

A thicker mélange leads to more homogenisation of water temperature in the upper 200 m of the fjord and warmer near-surface waters (Figure 8a). Both of these effects are due to increased upwelling from deeper icebergs with respect to the thinner mélange

profiles. A thicker mélange also leads to higher velocities at virtually all depths (Figure 8b). This is because the increase in





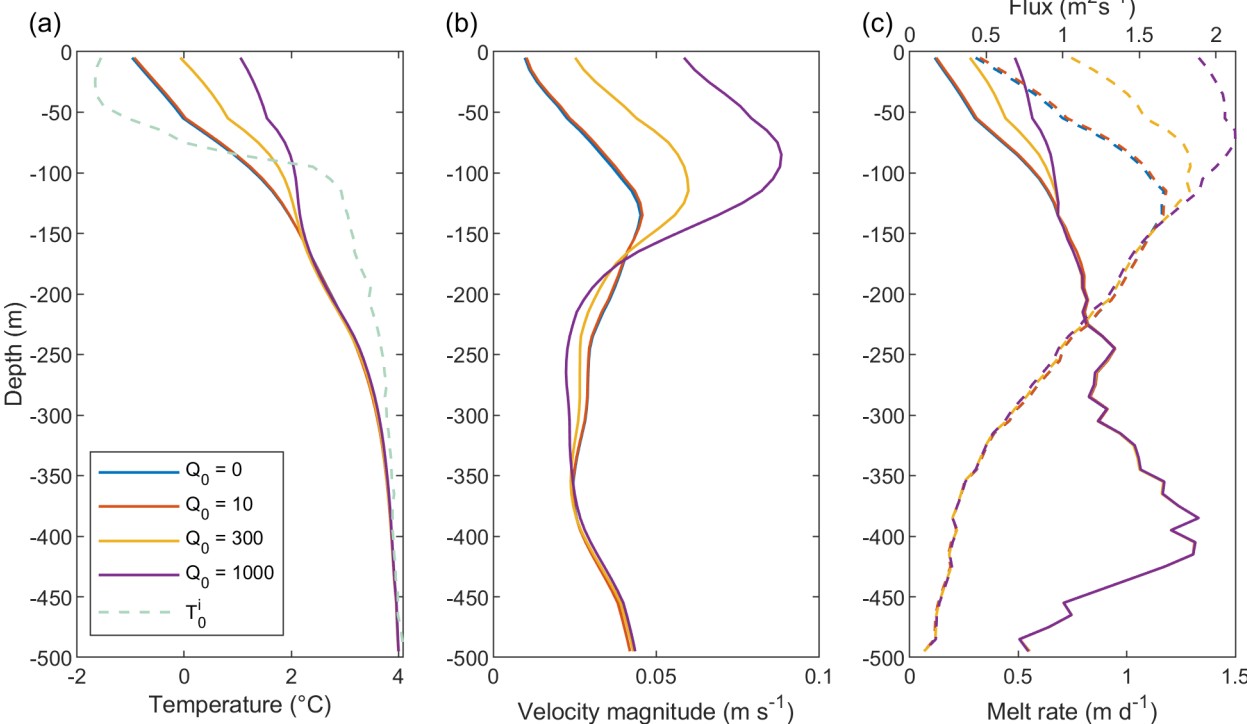

**Figure 6.** Sensitivity of ocean dynamics in the mélange to subglacial discharge flux. (a) Temperature. The dashed line shows the imposed boundary condition. (b) Velocity magnitude and (c) mélange melt rate (solid) and meltwater flux (dashed). All simulations have a temperature profile $T_0$, shown as the dashed line in (a), and a standard mélange thickness. All variables are averaged over the area of the mélange (the full width and depth of the fjord and 10 km down-fjord of the glacier terminus), as is the case for Figures 7 and 8.

upwelling generates a more vigorous circulation. The influence of mélange thickness on melt rates and meltwater fluxes is complex (Figure 8c), but in general melt rates respond to changes in the modelled temperature and velocity, with a thicker mélange having a higher melt rate near the surface due to high water temperatures and velocities in this region. The meltwater flux peaks at a depth of approximately 100 m for a thin mélange, but this flux is more spread out and peaks at a lower depth
for a thick mélange.

### 3.2.4  Overall melt rates and fluxes

For given ocean boundary conditions and mélange thickness profile, the sensitivity of overall mean mélange melt rate to subglacial discharge is sublinear in all cases (Figure 9a). Melt rates increase with subglacial discharge due to both the increased upwelling of warm waters (Figure 6a) and more vigorous circulation (Figure 6b), but the influence on overall melt rates remains
sublinear. The sensitivity of melt rates to deep water temperature is approximately linear (Figure 9a; see also section 3.2.5). This is primarily due to linear changes in the steady state temperature profile (Figure 7a), though water velocities do increase slightly



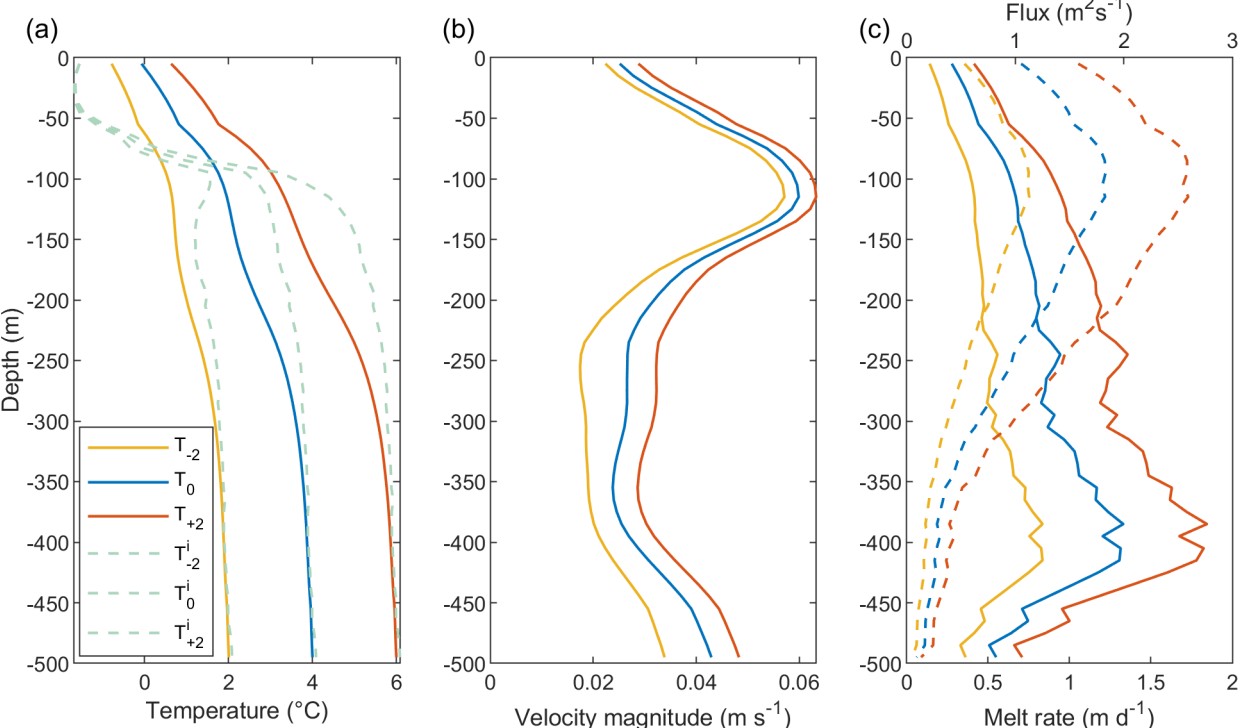

**Figure 7.** Sensitivity of ocean dynamics in the mélange to deep water temperature. (a) Temperature. The dashed lines show the imposed boundary conditions. (b) Velocity magnitude and (c) mélange melt rate (solid) and meltwater flux (dashed). All simulations have a subglacial discharge flux $Q_0 = 300 \text{ m}^3\text{s}^{-1}$ and a standard mélange thickness.

with higher temperatures (Figure 7b), contributing to increased melt rates. Mélange melt rates increase with mélange thickness in a non-linear fashion (Figure 9a): for higher values of subglacial discharge, thickening the mélange has less of an effect on the melt rate. This is because higher subglacial discharge flux corresponds to a more vigorous plume which homogenises the temperature of the water column (Figure 6a), resulting in shallow-drafted and deep-drafted icebergs experiencing similar thermal forcing and therefore melt rates.

The sensitivity of the total mélange meltwater flux to subglacial discharge and deep water temperature is similar to the sensitivity of the mean mélange melt rate to both of these variables (Figure 9b). This is because the meltwater flux is simply the melt rate scaled by the mélange surface area. However, the thick mélange, which is comprised of a smaller total number of (larger) icebergs, has a smaller submerged surface area. This means that, despite the fact that the meltwater flux per unit area (i.e. the melt rate) for the thick mélange is higher than for the standard mélange (Figure 9a), the total meltwater flux for the thick mélange is actually lower than the standard mélange meltwater flux (Figure 9b). The reverse is true for the thin mélange, which is comprised of a larger total number of (smaller) icebergs and so has a larger submerged surface area. Relatively speaking, the total meltwater flux for the thin mélange is higher because of its large submerged surface area.



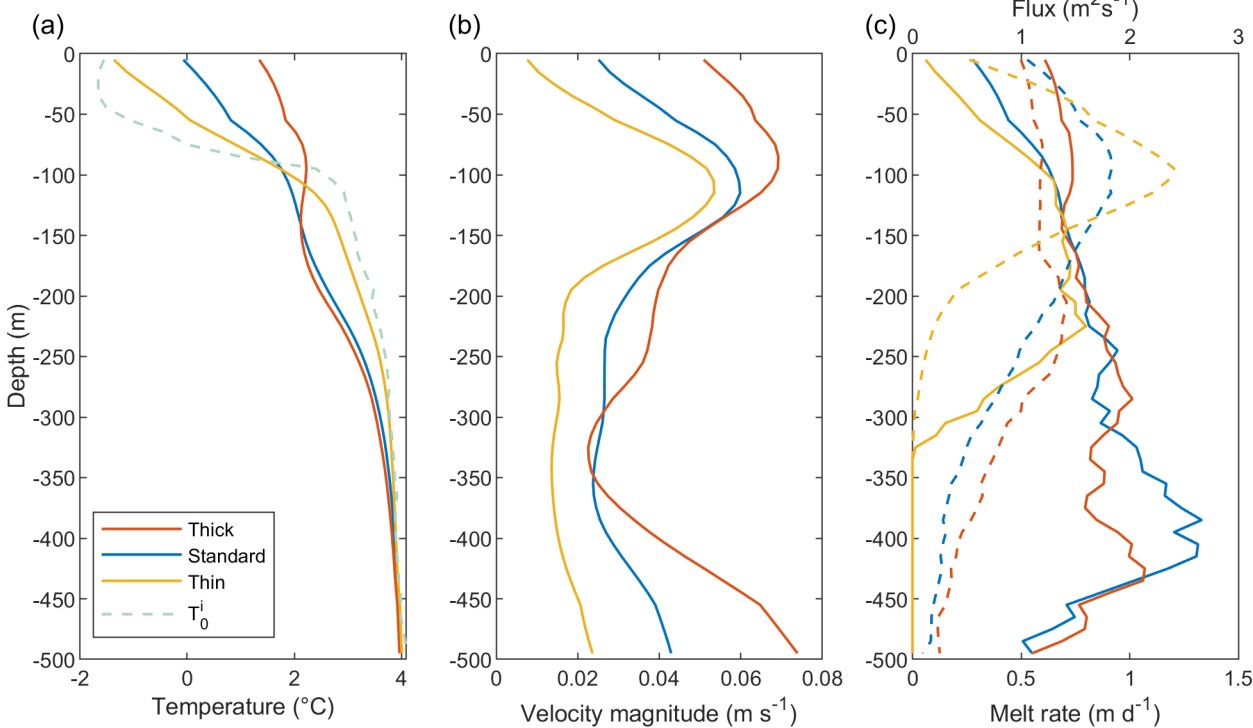

**Figure 8.** Sensitivity of ocean dynamics in the mélange to mélange thickness. (a) Temperature. The dashed line shows the imposed boundary condition. (b) Velocity magnitude and (c) mélange melt rate (solid) and meltwater flux (dashed). All simulations have a subglacial discharge flux of $Q_0 = 300 \, \mathrm{m^3 s^{-1}}$ and a temperature profile $T_0$.

### 3.2.5 Parameterisation

In light of the general trends discussed in section 3.2.4 as well as previous studies which have parameterised submarine melting (e.g., Rignot et al., 2016), we seek a generalised parameterisation of the mélange melt rate $m$ based on the simulations conducted in this study of the form

$$m = (a + bQ_0^c)\,\mathrm{TF}^d \tag{3}$$

where $a, b, c, d$ are constants, $Q_0$ is the subglacial discharge, and TF represents the oceanic thermal forcing. The thermal forcing as as function of depth $\mathrm{TF}(z)$ is given by

$$\mathrm{TF}(z) = T(z) - T_f(z) = T(z) - (\lambda_1 S(z) + \lambda_2 + \lambda_3 z) \tag{4}$$





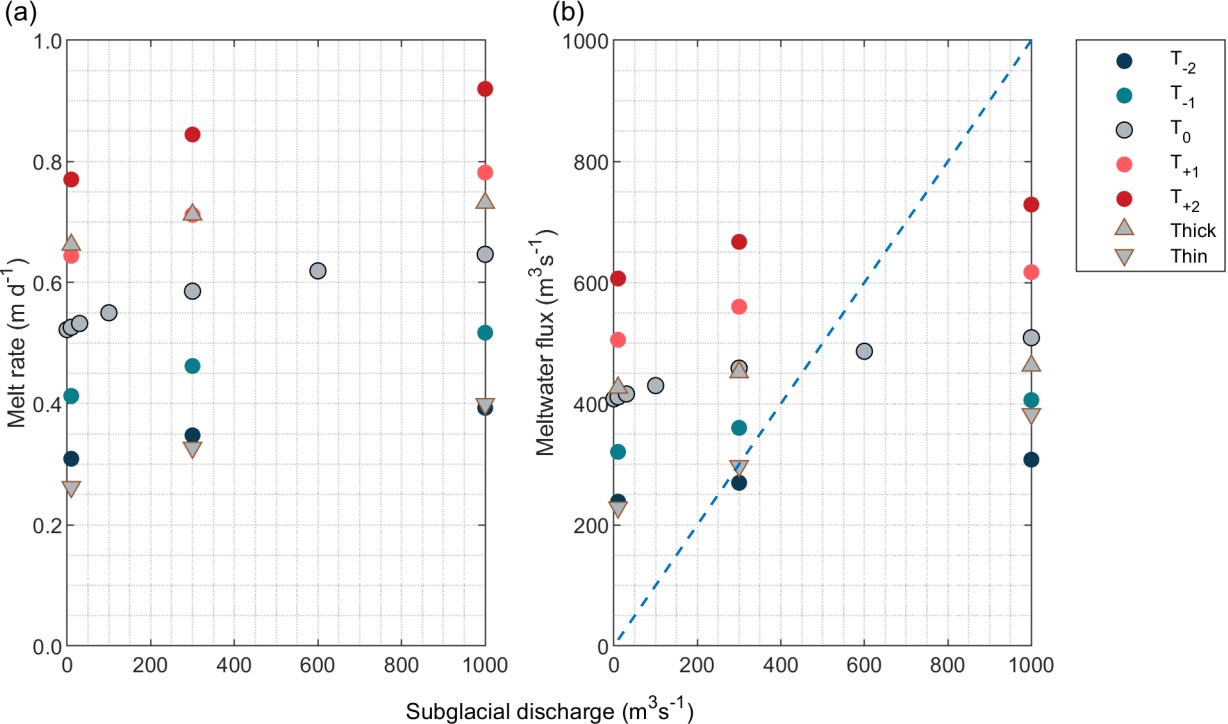

**Figure 9.** (a) The average melt rate and (b) total meltwater flux across all the simulations. The dashed blue line in (b) is the line for which the meltwater flux is equal to the subglacial discharge flux. The colour of the simulation refers to its boundary temperature profile, and the shape refers to its mélange thickness.

where $T(z)$ and $S(z)$ are the initial temperature and salinity profiles and the $\lambda_i$ are constants used to calculate the in-situ freezing point $T_f(z)$ (Jenkins, 2011). The thermal forcing which goes into the parameterisation of equation 3 is then the mean

of TF$(z)$ over a depth of $200 - 500$ m, a choice we made since these are the source waters that are upwelling to provide most of the heat that melts the mélange. The thermal forcing is defined as a function of the initial conditions $T(z)$ and $S(z)$ so that it can be calculated without the need to run any simulations.

On the whole, there is very solid agreement between simulated and parameterised melt rates (Figure 10). The parameterisation is given in equation 3, with values of $a = 7.30 \times 10^{-2}$, $b = 4.12 \times 10^{-4}$, $c = 5.45 \times 10^{-1}$ and $d = 1.16$. The mean melt

rates scale sublinearly with subglacial discharge (exponent 0.545) and slightly superlinearly with temperature (exponent 1.16). The value of $a = 0.073$ represents the background melting in the absence of any subglacial discharge, simply due to iceberg melting and the circulation which is set up as a result. Only the standard mélange thickness simulations have been used to generate this parameterisation.



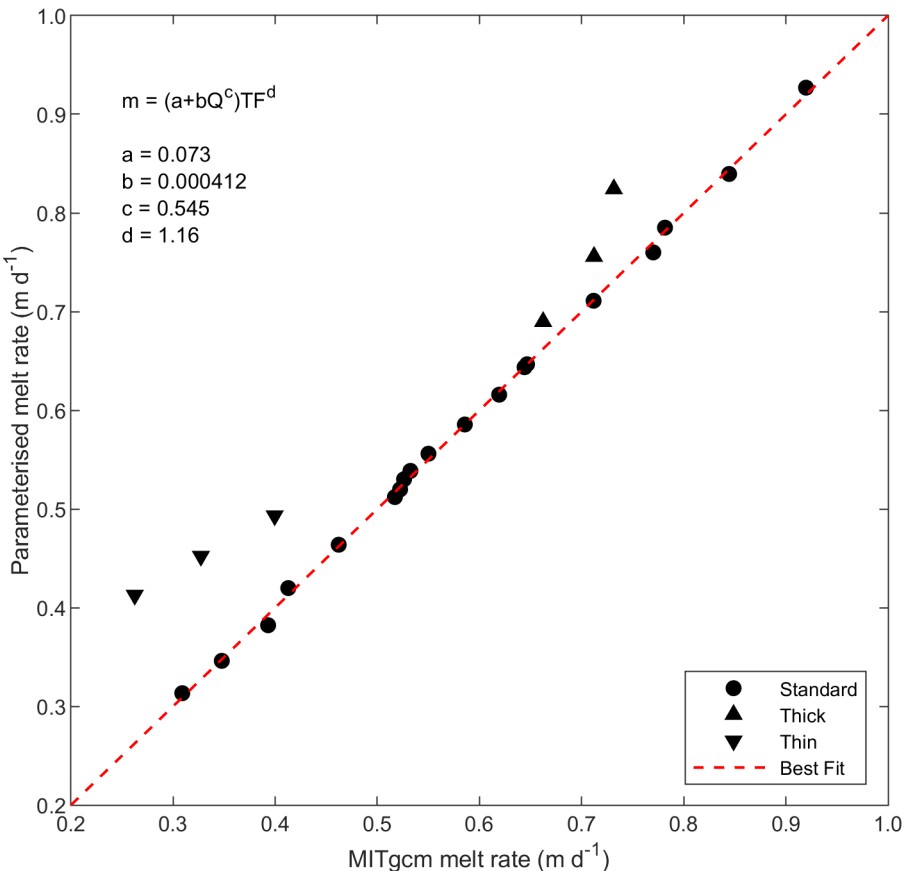

**Figure 10.** Parameterised melt rates compared with simulated melt rates. Simulations with standard mélange thickness are shown as circles; simulations with thick and thin mélange thickness are shown as upwards and downwards pointing triangles, respectively. Only the standard mélange thickness simulations are used to generate the parameterisation and best fit line.

## 4   Discussion

We have used MITgcm to model the circulation of ocean waters through a range of ice mélanges down-fjord from an idealised marine-terminating glacier system and estimate the resultant melt. The simulations suggest that warming always occurs in the top layer of the fjord, but that the magnitude of this warming grows with increased subglacial discharge, increased deep water temperature and increased mélange thickness. Similarly, water velocities increase – particularly in the upper $100\,\mathrm{m}$ of the fjord – with increased subglacial discharge, increased deep water temperature and increased mélange thickness. Since the

simulated melt rate is a function of water temperature and velocity, and the meltwater flux is essentially the melt rate scaled by the mélange surface area, melt rates and meltwater flux also increase sublinearly with subglacial discharge, superlinearly with ocean temperature and non-linearly with mélange thickness. The melt rate dependence on subglacial discharge flux and



ocean temperature can be summarised in a parameterisation with exponents of 0.55 and 1.16 respectively. Rignot et al. (2016) modelled the ocean melt of the calving front of multiple Greenlandic glaciers and obtained exponents of 0.39 and 1.18 for the sensitivities of the calving front melt rate on subglacial discharge and ocean temperature. This suggests that the ocean melt of ice mélange is broadly similar to the submarine melting of the calving front in terms of scalings with environmental forcings.

## 4.1 Parameterisation

The parameterisation we have developed in equation 3 is limited by the lack of reference to the mélange thickness. The sensitivity of the simulation to mélange thickness was explored in section 3.2.3. The influence of mélange thickness on modelled melt rates is complex: a thicker mélange leads to warmer near-surface waters (Figure 8a) and higher velocities at all depths (Figure 8b), but we were unable to find a simple way to include the mélange thickness in the melt rate parameterisation. In both the thick mélange and thin mélange cases, the parameterisation overestimates the simulated melt rate (Figure 10): on average, the melt rate for the thick mélange is $0.05 \mathrm{~m\,d^{-1}}$ too high and the melt rate for the thin mélange is $0.12 \mathrm{~m\,d^{-1}}$ too high. We would expect the thickness to contribute to the thermal forcing term, with a thicker ice mélange experiencing higher thermal forcing due to an increased submerged surface area of ice in contact with deep warm water. Incorporating the mélange thickness into the parameterisation is a key task in future work.

## 4.2 Buttressing force

One of the key motivations for studying ice mélange is to understand the impact mélange melting may have on glacier dynamics. Studies have shown that the buttressing force is highly sensitive to the submarine mélange melt rate (Amundson et al., 2024), meaning that an understanding of mélange melt rates is crucial to be able to accurately model the force balance at glacier termini.

Mélange thickness depends on the supply of icebergs into the mélange (i.e. the calving rate), the removal of icebergs from the mélange (i.e. the outflow into open fjord waters) and the melt rate of the mélange itself. Our results suggest that increased subglacial discharge and increased temperature of waters at depth in the fjord both result in higher mélange melt rates. Moreover, thicker mélange profiles, which extend deeper into warmer waters, experience significantly higher melt rates than thinner mélange profiles. An increase in the mélange melt rate will contribute to the mélange thinning over time, but the real relationship between melt rate and mélange thickness is complicated and depends on the iceberg flux into and out of the mélange. A potential extension of the current study would be to link mélange melt rates explicitly with calving rates at the glacier terminus via the buttressing force to gain a better understanding of the evolution of ice mélange. Observations of Sermeq Kujalleq by Joughin et al. (2020) suggest that the glacier responds to ocean forcing not through submarine melting of the glacier front but through the melting of its ice mélange and the associated variation in backstress. Our results here support the idea that the melting of an ice mélange would be sensitive to ocean temperature, and in particular to deep ocean temperature within the fjord.





### 4.3 Comparison with observations

Our results are broadly comparable with observations of mélange melting. Across all simulations, the value for the mean mélange melt rate varies between $0.26 - 0.92 \, \mathrm{m\,d^{-1}}$ and the total mélange meltwater flux varies between $228 - 729 \, \mathrm{m^3 s^{-1}}$. All observations discussed in this section studied Sermilik fjord, and so in theory should be directly comparable to our simulations.

Enderlin et al. (2016) use remote sensing at Sermilik fjord to find melt rates in the range of $0.05 - 0.67 \, \mathrm{m\,d^{-1}}$ and meltwater fluxes in the range of $126 - 494 \, \mathrm{m^3 s^{-1}}$. They find that melt rates for deep-drafted icebergs are three times larger than melt rates

for shallow icebergs and their observations cover the months of June, July and October. Our simulated mélange submerged area ranges from $65 \, \mathrm{km^2}$ (thick mélange) to $92 \, \mathrm{km^2}$ (thin mélange). This is slightly lower than the observations of Enderlin et al. (2016), who estimate that the area varies from $132 - 261 \, \mathrm{km^2}$. We note that our mélange length of $10 \, \mathrm{km}$ only focuses on the part of the mélange nearest to the glacier terminus and does not capture the full extent of the mélange, which may account for the underestimate in mélange submerged area.

Moon et al. (2018) use observations at Sermilik fjord to drive an iceberg-melt model which includes melting via wave erosion, forced and free convection in air and (depth-dependent) forced and free convection in water, and estimate the mélange meltwater flux to be approximately $200 - 350 \, \mathrm{m^3 s^{-1}}$ in the summer assuming that the mélange constitutes 30-40% of the total fjord iceberg melt. Moyer et al. (2019b) use Sentinel-2 imagery to analyse the region of Sermilik fjord outside of the ice mélange, but under the same assumption their estimate of the total meltwater flux between May and November is approximately

$450 - 1460 \, \mathrm{m^3 s^{-1}}$.

Our simulations concur with the result found elsewhere that the iceberg meltwater flux is a dominant source of freshwater in the winter under low subglacial discharge scenarios (Enderlin et al., 2016; Moon et al., 2018). This is demonstrated clearly in Figure 9b where the dashed line represents where the iceberg meltwater flux is equal to the subglacial discharge flux. For simulations above this line, the iceberg meltwater flux dominates; for simulations below this line, the subglacial discharge flux

dominates. Our results suggest that for low subglacial discharges (i.e. winter) and higher ocean temperatures, iceberg meltwater flux is more likely to dominate; for higher subglacial discharges (i.e. summer) and lower ocean temperatures, the subglacial discharge flux is more likely to be the dominant freshwater source in a fjord system.

### 4.4 Comparison with similar studies

A number of other studies have simulated fjords with ice mélange. Hughes (2022) simulated the flow around hundreds of

passive icebergs and found that the maximum average current speed was at or below the draft of the deepest icebergs. In this study, however, the maximum current speed was consistently around $100 \, \mathrm{m}$ depth. There are three key differences in our model setup compared with that of Hughes (2022). Firstly, Hughes (2022) uses a half-sinusoid forcing velocity with zero depth mean, imposing a velocity at the surface and the seafloor to represent the overall velocity structure in many of Greenland's fjords. In contrast, we prescribe a boundary condition to explicitly represent subglacial discharge and observe the general velocity

structure qualitatively (Figure 3a and Figures 6b, 7b, 8b). Secondly, Hughes (2022) generates iceberg drafts explicitly from observed iceberg distributions such as Sulak et al. (2017), whereas we generate our own iceberg distributions as described in



section 2.3. Qualitatively, the iceberg distribution of our standard mélange is similar (compare our Figure 2b to Figure 3 in Hughes (2022)), but the percentage of deeper-drafted icebergs will be different for the different mélange profiles. Thirdly, we include thermodynamic effects, including melt-driven convection, which have an influence on the fjord-wide circulation which is established in the simulation. Taken together, we suggest that the depth of maximum current speed seems to be relatively insensitive to the maximum iceberg depths in the fjord and is instead mainly a function of the buoyancy of the plume and the strength of the subglacial discharge flux (Figure 6b).

As discussed in section 1, Davison et al. (2020) approximates iceberg melting by implementing the 'IceBerg' package in MITgcm. This assigns an iceberg concentration to each grid box, estimates the resultant melting and then freshens, cools and accelerates the ocean water in each grid box accordingly. In contrast, we explicitly resolve the flow and melting around individual icebergs. Nevertheless, our results support the conclusions found in Davison et al. (2020) that the iceberg melt rate and meltwater flux scales with subglacial discharge and that iceberg melting is, on its own, a significant factor in the freshwater budget of a fjord and capable of driving its own fjord circulation.

## 4.5 Transferability to other systems

In terms of fjord dimensions, stratification and mélange thickness, our simulations have been designed to resemble Sermilik Fjord in southeast Greenland. We have tried to make our results more applicable to other systems by generalising the model setup and including different mélange thicknesses and temperature profiles. However, the overall shape of the temperature profile (in particular the temperature of the upper layer) is the same for all temperature anomalies, and all of our simulations have used a single salinity profile. Nevertheless, we expect our results to be applicable to systems that have similar characteristics to Sermilik fjord – that is, fjords of comparable dimensions with permanent ice mélange and a similar stratification of deep warm water underlying cooler water. We expect this to include other large glacier-fjord systems in Greenland, including Sermeq Kujalleq and Kangerlussuaq. The precise values in the melt rate parameterisation (equation 3) might vary, but the general concepts discussed above regarding the sensitivity of the mélange melt rate to subglacial discharge flux, deep ocean temperature and mélange thickness should carry over. A potential extension of the current study would be to repeat the simulations with the specific geometry and oceanographic conditions of various glacier-fjord systems around Greenland.

## 4.6 Limitations of the study

Uncertainty in the precise make-up of a mélange leads to uncertainty in the simulated melt rate. This uncertainty is primarily due to two factors: the mélange thickness and the distribution of icebergs in the mélange. Firstly, we have tried to create a realistic mélange by imposing a linearly decreasing mélange thickness. Our assumption that the mélange has a linearly decreasing profile is not unreasonable and is supported in some observational studies of mélange thickness, particularly in Sermilik Fjord (Enderlin et al., 2016). However, Burton et al. (2018) derive an expression for mélange thickness which decreases exponentially which depends on the coefficient of internal friction of the mélange. A potential extension of the current study would be to investigate the effect of different mélange profile shapes on the simulated melt rate.





Secondly, we randomly sample icebergs from artificial iceberg distributions to create an ice mélange. Whilst observations

look at the frequency of horizontal areas in an iceberg distribution pertaining to an entire ice mélange or fjord (Sulak et al., 2017; Shiggins et al., 2023), the method we have developed uses the frequency of iceberg drafts to generate five distinct iceberg distributions within the same ice mélange. This has the advantage of being able to generate a mélange with a realistic distribution of iceberg drafts as long as the mélange thickness is known or observed. However, this method assumes that it makes sense to have distinct iceberg distributions within the same mélange. We believe that, provided a given segment of the

mélange is sufficiently large to have enough icebergs to describe a distribution but sufficiently small that its mean iceberg draft is distinct from a separate segment, we would expect the iceberg populations of different segments of an ice mélange to be described by distinct power-law distributions. We chose to separate the mélange into five segments to attempt to achieve this balance between the segments having neither too many nor too few icebergs.

Iceberg shape is poorly represented in numerical models, which in turn leads to uncertainty in simulated melt rates. We

have treated icebergs as smooth cuboids, which is clearly a simplification of the true range of iceberg shape and surface roughness. On the larger scale, cuboids resemble true icebergs with their steep and sharp vertical faces (Hughes, 2022), though large-scale roughness would have the potential to influence large-scale circulation. If cuboid icebergs were instead replaced with smooth cylinders, water would find different pathways through the mélange, impacting the overall flow and, as a result, the simulated melt rates. On the smaller scale, a rough ice surface may increase the turbulence of the water flow, producing

ice scallops (Gilpin et al., 1980; Cenedese and Straneo, 2023). This can then further enhance the turbulent heat transfer to the ice, potentially increasing melt rates (Cenedese and Straneo, 2023). The impact of such surface details would have to be encapsulated in the melt rate parameterisation even in high-resolution studies.

The key limitation in this study is the uncertainty associated with the exact form of the three-equation melt rate parameterisation which leads directly to uncertainty in the simulated mélange melt rates. In particular, whilst the standard form of this

parameterisation uses a single value for the drag coefficient, in reality the drag experienced by fluid at the ice-ocean interface will depend both on the slope of the interface and the driver of the fluid flow itself. Zhao et al. (2024) found three specific flow regimes: plume-driven convection, melt-driven convection, and a general fjord-wide circulation. In light of this, we have tried to adapt the standard melt rate parameterisation (Holland and Jenkins, 1999; Jenkins, 2011) by using different values of the drag coefficient in different flow scenarios at the ice-ocean interface (Zhao et al., 2024). In particular, we have used thresholds

on the fluid velocity in a particular grid cell as a mask for the three distinct melt regimes (section 2.4). Despite this, converting a water velocity and temperature beside the ice to a generalised melt rate via a parameterisation is still limited by a lack of constraints on the values of parameters used in the parameterisation. More observational studies of iceberg melting are needed to decrease the uncertainty in these values.

In light of the above limitations, we have more confidence in the relative changes in mélange melt rate between different

simulations than we do on the absolute values of melt rate found in this study. In other words, the sensitivity of the mélange melt rate to subglacial discharge flux, deep ocean temperature and mélange thickness found in this study are results which we believe are independent of the precise values of the constants used in a melt rate parameterisation. Furthermore, we also note



that despite these potential limitations, our melt rate values are not dissimilar from those estimated in real ice mélange (section 4.3).

## 5   Conclusions

The melting of an ice mélange can have a significant impact on a glacier and its fjord by reducing the buttressing force on a glacier terminus and releasing a substantial freshwater flux. Inspired by the need for a systematic analysis of how the ocean melt rate of ice mélange varies under different environmental conditions, we have run high-resolution numerical simulations using MITgcm to model the flow of waters through an ice mélange close to idealised marine-terminating glacier systems around Greenland and estimated the resultant melt. Specifically, we chose a domain size and ice mélange configuration based on the Helheim Glacier – Sermilik Fjord system, but our simulations are intended to be representative of large Greenlandic glacier-fjord systems.

We found that near-surface waters within the ice mélange always warm due to the upwelling of deep warm waters, driven both by the subglacial discharge plume and by the melt-driven convection on the sides of the icebergs themselves, and that the magnitude of this warming grows with increased subglacial discharge, increased deep water temperature and increased mélange thickness. Moreover, the magnitude of the simulated mélange melt rates varies from $0.26 \ \mathrm{md}^{-1}$ to $0.92 \ \mathrm{md}^{-1}$ and is in good agreement with observational estimates.

Furthermore, we have developed a parameterisation of the mélange melt rate $m$ which depends on the subglacial discharge flux $Q_0$ and the thermal forcing TF and is in the form $m = (a + bQ_0^c)\mathrm{TF}^d$, with $a = 7.30 \times 10^{-2}$, $b = 4.12 \times 10^{-4}$, $c = 5.45 \times 10^{-1}$ and $d = 1.16$. Our results highlight that mélange melt rates increase sublinearly with subglacial discharge and approximately linearly with ocean temperature, suggesting that ocean melting of ice mélange follows similar environmental sensitivities as submarine melting at glacier calving fronts. The complex response of the melt rate to the thickness of an ice mélange meant that we were unable to incorporate the mélange thickness into this parameterisation, and doing so is a key task for future work. Nevertheless, this study is a step towards a better understanding of the sensitivity of mélange melting to changing environmental conditions – a crucial process to model accurately if we want to predict how Greenland's glaciers and fjords will change in the future.

## Appendix A: Generating an iceberg distribution

All modelled icebergs in this study are cuboid to simplify simulations, but in reality icebergs have infinitely many shapes. Nevertheless, assuming cuboid icebergs, there is a simple relationship between an iceberg's horizontal area $A$, total volume $V$ and draft $H$: $V = AH \times (\rho_w/\rho_i)$, where $\rho_w$ and $\rho_i$ are the densities of water and ice respectively. Since iceberg area is related to iceberg volume, and iceberg volume to iceberg draft for cuboid icebergs, in addition to the power law distribution $N \propto A^{-\alpha}$, we expect there to be a similar power law distribution of the form $N \propto H^{-\beta}$. That is, shallower icebergs should be much more common than deeper ones. Populating an artificial ice mélange by randomly sampling from an iceberg distribution of the form




$N \propto H^{-\beta}$ is one way of respecting the fact that icebergs must replicate observed iceberg size power law distributions and have
realistic aspect ratios as discussed in the main text.

Consider a power law distribution for the number of icebergs $N$ of draft $H$ of the form $N(H) = \eta H^{-\beta}$ with undetermined
constants $\eta$ and $\beta$. For this to be a probability distribution, we require $\int_{H_{\min}}^{H_{\max}} N(H) = 1$ where the parameters $H_{\min}$ and $H_{\max}$
represent the smallest and largest iceberg draft in the distribution. We set $H_{\max}$ to be the full depth of the fjord – this represents
the possibility of full-depth calving at the calving front. To determine $H_{\min}$, we combine $V = AH \times (\rho_w/\rho_i)$ and $V = aA^b$ to
give an explicit relationship for iceberg draft $H$ in terms of iceberg area $A$: $H = (\rho_i/\rho_w) \times aA^{b-1}$. There is a limit on the size
of $A$ determined by model resolution ($A_{\min} = \mathrm{dx} \times \mathrm{dy}$), and so $H_{\min}$ is set by $H_{\min} = (\rho_i/\rho_w) \times aA_{\min}^{b-1}$.

Applying the normalisation condition fixes the value of $\eta$ and leaves only one undetermined constant $\beta$:

$$N(H) = \left( \frac{-\beta+1}{H_{\max}^{-\beta+1} - H_{\min}^{-\beta+1}} \right) H^{-\beta} \tag{A1}$$

This is the same as equation 2 in the main text. The mean $\bar{N}$ of this distribution is then

$$\bar{N} = \frac{\int_{H_{\min}}^{H_{\max}} H N(H) dH}{\int_{H_{\min}}^{H_{\max}} N(H) dH} = \frac{-\beta+1}{-\beta+2} \times \frac{H_{\max}^{-\beta+2} - H_{\min}^{-\beta+2}}{H_{\max}^{-\beta+1} - H_{\min}^{-\beta+1}} \tag{A2}$$

$\bar{N} = \bar{N}(\beta, H_{\min}, H_{\max})$, i.e. $\bar{N}$ is a function of $\beta$, $H_{\min}$ and $H_{\max}$. $H_{\min}$ and $H_{\max}$ have been fixed above. $\bar{N}$ is set by
observations of mean mélange thickness and then equation A2 is used to determine $\beta$. In more detail: we split the domain
into equal-width segments down-fjord, and pick a value mean mélange thickness $\bar{N}$ in each segment, based on observational
data (e.g., Enderlin et al., 2016). We then numerically solve equation A2 for $\beta$ in each segment. This then provides a unique
probability distribution of iceberg drafts $N(H)$ for each segment, with the mean of each probability distribution being the
target mean draft of icebergs in that segment.

## Appendix B: Inverse transform sampling

Inverse transform sampling is a way of randomly sampling from an arbitrary probability distribution. First, we make a cumu-
lative distribution function $F(H)$ using the definition of $N(H)$ in equation 2:

$$F(H) = \int_{H_{\min}}^{H} N(t) dt = \frac{H^{-\beta+1} - H_{\min}^{-\beta+1}}{H_{\max}^{-\beta+1} - H_{\min}^{-\beta+1}} \tag{B1}$$

Next, we find the inverse $\tilde{F}(H)$:

$$\tilde{F}(H) = \left[ H \left( H_{\max}^{-\beta+1} - H_{\min}^{-\beta+1} \right) + H_{\min}^{-\beta+1} \right]^{\frac{1}{-\beta+1}} \tag{B2}$$





To randomly sample an iceberg draft from our iceberg distribution, we generate a random number $u$ such that $0 < u < 1$; a random iceberg draft $h$ is then given by $h = \tilde{F}(u)$.





**Appendix C: Table of simulations**

**Table C1.** All the simulations run in this study. The naming convention is: mélange-thickness_subglacial-discharge-flux_temperature-profile, where 'ms', 'mk' and 'mn' stand for a standard, thick and thin mélange thickness respectively. The mélange geometry is given in terms of mean iceberg draft at the glacier terminus $d_a$ and mean iceberg draft at the end of the mélange $d_b$. For the standard mélange thickness and the standard temperature profile $T_0$, all subglacial discharge values were run. For all other temperature profiles, only values of $Q_0 = 10, 300, 1000 \, \mathrm{m^3 s^{-1}}$ were chosen to limit the number of simulations. For the thick and thin mélange profiles, only the standard temperature profile was run (with $Q_0 = 10, 300, 1000 \, \mathrm{m^3 s^{-1}}$) for the same reason.

| Simulation name | Mélange geometry $[d_a, d_b]$ (m) | $Q_0$ $(m^3 s^{-1})$ | Temperature profile |
|---|---|---|---|
| ms_q0_t0 | [100,50] | 10 | $T_0$ |
| ms_q10_t0 | [100,50] | 10 | $T_0$ |
| ms_q30_t0 | [100,50] | 30 | $T_0$ |
| ms_q100_t0 | [100,50] | 100 | $T_0$ |
| ms_q300_t0 | [100,50] | 300 | $T_0$ |
| ms_q600_t0 | [100,50] | 600 | $T_0$ |
| ms_q1000_t0 | [100,50] | 1000 | $T_0$ |
| ms_q10_t1 | [100,50] | 10 | $T_1$ |
| ms_q300_t1 | [100,50] | 300 | $T_1$ |
| ms_q1000_t1 | [100,50] | 1000 | $T_1$ |
| ms_q10_t-1 | [100,50] | 10 | $T_{-1}$ |
| ms_q300_t-1 | [100,50] | 300 | $T_{-1}$ |
| ms_q1000_t-1 | [100,50] | 1000 | $T_{-1}$ |
| ms_q10_t2 | [100,50] | 10 | $T_2$ |
| ms_q300_t2 | [100,50] | 300 | $T_2$ |
| ms_q1000_t2 | [100,50] | 1000 | $T_2$ |
| ms_q10_t-2 | [100,50] | 10 | $T_{-2}$ |
| ms_q300_t-2 | [100,50] | 300 | $T_{-2}$ |
| ms_q1000_t-2 | [100,50] | 1000 | $T_{-2}$ |
| mk_q10_t0 | [200,100] | 10 | $T_0$ |
| mk_q300_t0 | [200,100] | 300 | $T_0$ |
| mk_q1000_t0 | [200,100] | 1000 | $T_0$ |
| mn_q10_t0 | [50,25] | 10 | $T_0$ |
| mn_q300_t0 | [50,25] | 300 | $T_0$ |
| mn_q1000_t0 | [50,25] | 1000 | $T_0$ |



*Code and data availability.* The archive accessible at https://doi.org/10.5281/zenodo.14531115 (Jain, 2024) includes the code needed to generate the ice mélange profiles, the code needed to run the MITgcm simulations, and the output of the last timestep of the ms_q300_t0 simulation.

*Author contributions.* LJ, DAS and PN designed the research. LJ set up the MITgcm runs with help from DAS. LJ ran the simulations and analysed the results. LJ wrote the paper, with input from all other authors.

*Competing interests.* No competing interests are present.

*Acknowledgements.* LJ is funded by NERC through the E4 DTP studentship NE/S007407/1. DAS acknowledges support from NERC Independent Research Fellowship NE/T011920/1.



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
