# Peer review of "Modelling ocean melt of ice mélange at Greenland's marine-terminating glaciers"

_EGUsphere, 2024_

## Referee Comment (RC1)

Review of "Modelling ocean melt of ice melange at Greenland's marine-terminating glaciers"

Jain et al.

Date: 20 Feb 2025

Assessment: Major revision

Reviewed by: Erwin Lambert

The authors have configured an idealised model setup for a Greenland fjord filled with melange. Based on several backed up constraints, they have created a distribution of icebergs and added these to their MITgcm setup. They have analysed the impact of melange melt on the fjord circulation and heat distribution, and the sensitivity thereof to temperature forcing, subglacial melt, and melange thickness. They have expressed the sensitivity to temperature forcing and subglacial melt in a parameterisation, which may apply to melange-filled fjord systems more generally.

The study is very relevant as the community is actively developing methods to represent climate forcing of the Greenland ice sheet. The ocean-driven melt of melange, through buttressing, is an important aspect of this. The study is generally well-designed with an extensive sensitivity analysis and a thoroughly constructed iceberg distribution. The paper overall was a pleasure to read and the figures clearly convey their message.

I do have several concerns with the current manuscript and therefore recommend a major revision before publication. The major concerns are threefold:

1) The choice of iceberg areal fraction lambda
    a. The authors back up their choice of 0.6 in lines 148-154. However, I believe a mistake was made in their calculation, and a maximum iceberg-to-sea ice ratio of 0.35 should imply a maximum total iceberg fraction of 0.26. A 'best estimate' between 0 and 0.26 should then be 0.13, a factor four smaller than what the authors used. I think the authors should reconsider their choice of lambda. If it is unfeasible to redo the simulations with a lower value, they should extensively discuss the impact of this choice on their results and conclusions, ideally backed up with a sensitivity run.
    b. I was surprised not to see lambda back in the Discussion. The authors discuss various sources of uncertainty, without picking up this one. I think this deserves some additional discussion.

c. Following on that, the authors should reconsider how valuable their comparison to observations of total subglacial meltwater fluxes (Sec 4.3) is considering this uncertainty. If indeed a lambda of 0.13 is more representative, I imagine the simulated meltwater flux should reduce by a factor four. The authors should critically consider which observational values are valid to compare to (and if there are none, so be it).

2) The description of the Results
   a. I see a few obvious patterns in the figures that the authors do not discuss, which I think is a shame. The manuscript would benefit from pointing out these results and analysing them.
      i. The asymmetry in Fig. 4b
      ii. The asymmetry in melt rate in Fig. 5b
      iii. The spatial correlation in Fig. 5b, which is absent in Fig. 5a
   b. The paragraphs in 3.2.1-3.2.3 very neatly describe one subpanel each. To my taste, this is a bit too neat, making me lose sight of the larger picture and the connection between the figures/sections. I suggest that the authors invest in a bit more inter-linkage here and perhaps step away from the 1 paragraph per subfigure. For example, in l. 330-331, the same process is pointed out as before. Make this explicit: "Similar to an enhanced subglacial meltwater flux and a warmer ambient ocean, a thicker melange increases upwelling and generates a more vigorous circulation. Hence, each of these processes causes an increase in upper-layer velocities".

3) The discussion: contents and structure
   a. You end with 'limitations of this study'. Depends on your taste, but I'd put this first, and then end with the good stuff (comparison and implications).
   b. Sec 4.2 (Buttressing) appears out of place. It is unclear how the new insights gained in this study reflect on buttressing. If they do, mention this explicitly, otherwise, this topic should be treated as motivation and be moved to the introduction.
   c. In Sec 4.3, the authors compare their submerged areas to observations. However, this is model input, not output. Hence, this comparison should be transferred to the methods section. Also, the authors should consider whether comparing absolute values in area makes sense considering the uncertainty in lambda.
   d. This study would benefit from more focus on 'lessons learned' in the discussion. As an example, the ISMIP7 team is currently developing protocols for the ocean forcing of the Greenland ice sheet. What guidance can you give that team based on your results? And if they cannot directly use your parameterisation, what additional research steps are necessary to do this?

Minor comments:

Abstract. Half of the abstract is intro/motivation, the rest mostly methodology. Please put more emphasis on the results/conclusions.

l. 55. Add explicitly that this impact (melange melt and not buttressing) is the topic of this paper.

l. 59. Mention 'due to entrainment'

l. 68. .. how **the** meltwater flux from icebergs influence**s the** large-scale ....

l. 75, 77. 'Exactly' and 'precisely' are unreasonable expectations for a model. Change to 'accurately' or similar.

l. 81 'This study' -> 'that study' (avoid ambiguity)

l. 120 Mention explicitly that the salinity profile is kept constant for all runs (this is only mentioned somewhere at the end of the manuscript)

Fig 1.b Make the colours in your different figures consistent. Here: use the same colours as in Fig 9.

l. 135 'constituted' -> 'constitute'

Eq 2. Becomes a bit clearer when changing –beta+1 to 1-beta

Fig 2.c Again, try to use the same colours and markers as in Figs 9 and/or 10

l. 220 No calving front melting in the analysis. But is it included in the simulation? (So an extra buoyancy source)? Please state this explicitly

l. 229 'simulate' -> 'simulating'

l. 232 **an** approximate steady state

l. 253 has been lost -> is lost

l. 272 **The** temperature

l. 274 mid-depths -> mid-depth

Fig 4b: xlabel should be 'across fjord'

Fig 4c: I'd replace 'velocity magnitude' with 'speed'. Velocity implies that it can be negative

l. 282 **a** considerable fluctuation; **a** variation

l. 283 a sum -> the sum

Fig 5: It is unclear why Fig 4 is smoothed, and Fig 5 is not. Is there a particular interest in the fine-scale variations? If not, I'd opt for consistency across the figures. Particularly in Fig 5b, a smoothed curve would better visualise whether there is an asymmetry in the meltwater flux or not

Fig 6: The near-overlap of blue and red curves makes them indistinguishable from the purple one. I suggest swapping orange and red. This also makes the order of the colours more intuitive.

l. 302 Is the shallower depth of neutral buoyancy determined by a fresher plume, a warmer plume, or significantly impacted by both?

l. 311-317 It's a bit difficult to follow the reasoning, while the point you try to make is quite straightforward. Please rewrite and to more directly get to the point.

l. 336 profile -> profiles

l. 339 This sentence implies that sublinear means a relatively weak sensitivity. This is not per se the case.

l. 340 Can you explain why this sensitivity is sublinear (what is the negative feedback / saturation process that decreases the sensitivity?)

l. 340 'Approximately linear'. Elsewhere, this is referred to as supralinear, which you indeed quantify in the parameterisation. Here, you give the explanation for the positive feedback inducing the supralinearity. Please link these statements and explanations together explicitly and make sure that what you interpret as supralinear is referred to as supralinear throughout the paper.

l. 344 **a** higher subglacial discharge

Fig 7: The caption isn't fully self-explanatory. Please explain that all variables are averaged horizontally over the full domain. And again, I'd suggest changing 'velocity magnitude' to speed (same for Fig 8)

l. 361 'as as' -> 'as a'

l. 363-367 I think it's more logical to say that T and S are the 'restoring / forcing' temperature and salinity. Because you state earlier that the results are independent of the initial conditions. The fact that you choose to initialise with the restoring conditions is irrelevant.

l. 363 You have a double use for lambda in the paper. I'd keep this here (usual nomenclature for the freezing point parameters) and change lambda for the iceberg area fraction. (Unless that's also a common usage, in that case, ignore this point)

l. 412 **the** deep ocean temperature

l. 419 'three times larger'. How does this compare to your results? And if they differ, can you explain why? If you don't compare this to your own results, it's not a relevant result to mention in your discussion.

Sec 4.3 You explicitly mention the seasonality in observations. How should I as a reader interpret this? Can your results be compared to the seasonally biased

observations? Or are your results representative of annual means? Please provide a bit of perspective

l. 431-437 These discussion points are again (strongly) dependent on the choice of lambda. Reconsider how these points should be perceived considering the uncertainty in lambda.

l. 440 this study -> our study (avoid ambiguity)

l. 442 'uses': this is unclear. Do they prescribe velocities, in which case the result (maximum at 100m) is trivial? Or do they restore velocities to this half-sinusoid, in which case the velocities are actually resolved? Please be specific

l. 467 'The precise values … might vary'. If you link the values to physical interpretation (see point above), you may be able to provide useful constraints: that, based on physical reasoning, c will generally be < 1, and d >1.

l. 478 Can you, based on the sensitivity to melange thickness, and your analysis of down-fjord gradients in melting, provide some insight into how important you think the exact distribution may be?

l. 473 (Sorry for going on about this the whole time) A third source of uncertainty is the total areal fraction of icebergs. This deserves a mention here.

l. 498 Are you sure this is 'the key limitation/uncertainty'? If so, why did you not do a sensitivity test to this? The simplest reply to this point is to change this to 'a key limitation/uncertainty', but perhaps, based on your knowledge and trial and error, you can provide a bit more insight than that on how important this aspect is.

l. 510 This is a contradictory sentence. The constants define the sensitivity, so the sensitivity cannot be independent of these constants.

l. 531 Here, it's again 'approximately linear' to ocean temperature. Make sure your own interpretation is consistent throughout.

l. 535 predict -> project

---

## Author Response (AR1)

**Introduction**

We thank the reviewers for their time and helpful comments. We are very pleased that the manuscript was well-received overall: the study is "very relevant", "generally well-designed" (reviewer 1) and "well thought-out" (reviewer 2) and the paper was "a pleasure to read" (reviewer 1) and "easy to follow" (reviewer 2). We have worked hard to address the reviewers' comments in the revised manuscript, and we respond to all comments below. Reviewer comments are in black; our comments are in blue. References to the line numbers in the original manuscript are styled O.X and references to line numbers in the revised manuscript are L.X.

**Reviewer 1**

**Major comments**

1. The choice of iceberg areal fraction lambda

The authors back up their choice of 0.6 in lines 148-154. However, I believe a mistake was made in their calculation, and a maximum iceberg-to-sea ice ratio of 0.35 should imply a maximum total iceberg fraction of 0.26. A 'best estimate' between 0 and 0.26 should then be 0.13, a factor four smaller than what the authors used. I think the authors should reconsider their choice of lambda. If it is unfeasible to redo the simulations with a lower value, they should extensively discuss the impact of this choice on their results and conclusions, ideally backed up with a sensitivity run.

We thank reviewer 1 for raising the choice of the iceberg areal fraction lambda. We agree that a maximum iceberg-to-sea ice ratio of 0.35 (Foga et al., 2014) should imply a maximum total iceberg fraction of 0.26, which is significantly lower than the value of 0.6 used in our paper. Due to the high computational expense of running the simulations, we are unfortunately unable to rerun all the simulations, but we can – as suggested by the reviewer – extensively address the sensitivity to lambda.

To address this, we have added four sensitivity experiments that consider lower values of lambda ($\lambda = 0.5, 0.4, 0.3, 0.2$) with a subglacial discharge of $Q_0 = 300 \text{ m}^3\text{s}^{-1}$, and a further two experiments to examine the effect that varying subglacial discharge flux has on melt rates and meltwater flux at a lower value of lambda ($\lambda = 0.2, Q_0 = 10, 1000 \text{ m}^3\text{s}^{-1}$). The four sensitivity experiments are discussed in a new section 3.2.4 and all simulations are included in the new Figure 10. These extra simulations demonstrate that the mélange melt rate is somewhat sensitive to lambda; melt rate varies by about 0.05 md[-1] when lambda is varied from 0.6 to 0.2. This is comparable to the variation in melt rate due to changes in subglacial discharge flux, but smaller than the variation due to changes in the temperature profile or the mélange thickness. Overall, changing $\lambda$ is therefore a small effect and the results from the originally submitted paper still hold. We have also included additional discussion concerning the value of $\lambda$ as requested (L.454-462).

I was surprised not to see lambda back in the Discussion. The authors discuss various sources of uncertainty, without picking up this one. I think this deserves some additional discussion.

We agree that lambda deserves additional discussion: we have added a paragraph in the "Limitations of study" section to discuss our method for choosing lambda and its effect on our results. This can be found in L.454-462.

Following on that, the authors should reconsider how valuable their comparison to observations of total subglacial meltwater fluxes (Sec 4.3) is considering this uncertainty. If indeed a lambda of 0.13 is more representative, I imagine the simulated meltwater flux should reduce by a factor four. The authors should critically consider which observational values are valid to compare to (and if there are none, so be it).

Given our additional simulations considering lower values of $\lambda$, we believe that our comparison to observations of total mélange meltwater fluxes (now Section 4.4) is still valuable. Although the meltwater flux is approximately linearly sensitive to lambda, the range of meltwater fluxes simulated in the study has not changed much because of the strong sensitivities to temperature and thickness that were already considered. We have adjusted the values of simulated total meltwater flux to encompass the full range of simulations, including those with lower values of $\lambda$, and modified the text accordingly (L.526).

2. The description of the Results

I see a few obvious patterns in the figures that the authors do not discuss, which I think is a shame. The manuscript would benefit from pointing out these results and analysing them.

i. The asymmetry in Fig. 4b
ii. The asymmetry in melt rate in Fig. 5b
iii. The spatial correlation in Fig. 5b, which is absent in Fig. 5a

We thank the reviewer for pointing out these additional patterns in our results which we have not discussed. We have addressed all of these comments as follows:

i. The asymmetry in Fig. 4b (whereby there is a higher water velocity and temperature on the northern side of the fjord) suggests that the subglacial discharge plume is primarily travelling along the northern edge of the domain, most likely due to the specific configuration of icebergs in the mélange. This is now addressed in L.284-287.

ii. The asymmetry in the melt rate in Fig. 5b (whereby the melt rate is higher in the northern half of the domain) is likely due to the same reason noted above of the plume preferentially travelling in this section of the fjord. This is now addressed in L.301-303.

iii. The spatial correlation between the melt rate and the meltwater flux in Fig. 5b

(across the fjord, averaged along the fjord) which is absent in Fig. 5a (along the fjord, averaged across the fjord) can be understood by considering what drives the variability in melt rate. The primary variability in melt rate occurs along the fjord – further away from the terminus, decreasing ocean temperature and velocity combined with a thinner mélange lead to lower melt rates. However, in Fig. 5b, this along-fjord variation is averaged, which allows the second-order variation to come through. This second order variation in melt rate arises due to the position of the plume (see point ii. above), which also influences the meltwater flux. Moreover, the short length-scale variation is due to the presence or absence of large, deep icebergs. These are submerged in deeper, warmer water, and so experience a higher melt rate and produce a higher meltwater flux. This is now addressed in L.312-318.

The paragraphs in 3.2.1-3.2.3 very neatly describe one subpanel each. To my taste, this is a bit too neat, making me lose sight of the larger picture and the connection between the figures/sections. I suggest that the authors invest in a bit more inter-linkage here and perhaps step away from the 1 paragraph per subfigure. For example, in O.330-331, the same process is pointed out as before. Make this explicit: "Similar to an enhanced subglacial meltwater flux and a warmer ambient ocean, a thicker mélange increases upwelling and generates a more vigorous circulation. Hence, each of these processes causes an increase in upper-layer velocities".

We have taken the reviewer's explicit example with regards to O.330-331 to interlink the description in 3.2.3 with previous sections (L.358-359). With regards to the one paragraph per subfigure structure, we feel that this neat and simple structure works well in this context: we first consider the sensitivity to each parameter in turn, and then broaden our perspective in section 3.2.5 when we consider overall melt rates and fluxes.

3. The discussion: contents and structure

You end with 'limitations of this study'. Depends on your taste, but I'd put this first, and then end with the good stuff (comparison and implications).

Thank you for the suggestion, we have done this.

Sec 4.2 (Buttressing) appears out of place. It is unclear how the new insights gained in this study reflect on buttressing. If they do, mention this explicitly, otherwise, this topic should be treated as motivation and be moved to the introduction.

We feel that our study does reflect on buttressing indirectly, but in a critical way, because mélange melt rate is a key control on mélange thickness, which in turn determines buttressing force. We have made this clearer in L.505. We have also incorporated some comments regarding "lessons learned" as suggested by the reviewer into this section (L.507-511).

In Sec 4.3, the authors compare their submerged areas to observations. However, this is model input, not output. Hence, this comparison should be transferred to the methods section. Also, the authors should consider whether comparing absolute values in area makes sense considering the uncertainty in lambda.

*Thanks, we have now moved this to the methods section (L.178-182). We have also briefly added a discussion of the submerged area of the $\lambda = 0.2$ simulation (which as expected is much smaller).*

This study would benefit from more focus on 'lessons learned' in the discussion. As an example, the ISMIP7 team is currently developing protocols for the ocean forcing of the Greenland ice sheet. What guidance can you give that team based on your results? And if they cannot directly use your parameterisation, what additional research steps are necessary to do this?

*We have added some explanation of the additional research steps which would be necessary to convert the parameterisation into useful ocean forcing data for the Greenland ice sheet into Section 4.3 (L.507-511).*

**Minor comments**

Abstract. Half of the abstract is intro/motivation, the rest mostly methodology. Please put more emphasis on the results/conclusions.

*Thanks for the suggestion – we have added more emphasis on the results and conclusions into the abstract, noting the influence of the mélange on the temperature stratification and the magnitude of modelled melt rates.*

O.55 Add explicitly that this impact (mélange melt and not buttressing) is the topic of this paper.

*Added in L.58.*

O.59 Mention 'due to entrainment'

*We believe that this statement regarding the conclusions of Hughes (2024) paper was not directly relevant to the logic of this paragraph and so we have deleted this sentence.*

O.68 … how **the** meltwater flux from icebergs influence**s the** large-scale …

*Changed in L.65-66.*

O.75, O.77. 'Exactly' and 'precisely' are unreasonable expectations for a model. Change to 'accurately' or similar.

*Point taken – changed to 'accurately' (L.74) and 'in detail' (L.76).*

O.81 'This study' -> 'that study' (avoid ambiguity)

*Changed in L.81.*

O.120 Mention explicitly that the salinity profile is kept constant for all runs (this is only mentioned somewhere at the end of the manuscript)

*We have added a sentence to mention this explicitly in L.119.*

Fig 1.b Make the colours in your different figures consistent. Here: use the same colours

as in Fig 9.

Thanks – we have changed the colour scheme in Fig. 1b accordingly.

O.135 'constituted' -> 'constitute'

Changed in L.136.

Eq 2. Becomes a bit clearer when changing –beta+1 to 1-beta

We have changed this in the main text and throughout the appendices.

Fig 2.c Again, try to use the same colours and markers as in Figs 9 and/or 10

Thanks – we have changed the markers in Fig. 2c accordingly.

O.220 No calving front melting in the analysis. But is it included in the simulation? (So an extra buoyancy source)? Please state this explicitly

We have added an explicit statement that melting of the calving front is included in the simulations in L.233.

O.229 'simulate' -> 'simulating'

This section has been rewritten to explain more clearly the set of simulations run.

O.232 **an** approximate steady state

Added "an" in L.250.

O.253 has been lost -> is lost

Changed in L.270.

O.272 **The** temperature

Added "the" in L.292.

O.274 mid-depths -> mid-depth

Changed in L.294.

Fig 4b: xlabel should be 'across fjord'

Thanks, this has been changed.

Fig 4c: I'd replace 'velocity magnitude' with 'speed'. Velocity implies that it can be negative

Thanks for this suggestion – we have changed 'velocity magnitude' to 'speed' in all appropriate figures.

O.282 **a** considerable fluctuation; **a** variation

Added "a" in both instances in L.305.

O.283 a sum -> the sum

Changed in L.306.

Fig 5: It is unclear why Fig 4 is smoothed, and Fig 5 is not. Is there a particular interest in the fine-scale variations? If not, I'd opt for consistency across the figures. Particularly in Fig 5b, a smoothed curve would better visualise whether there is an asymmetry in the meltwater flux or not

Apologies – Fig. 5 was in fact smoothed in exactly the same way as Fig. 4. This has now been made explicit in the Figure caption. We agree that it is important to adopt consistency across the figures.

Fig 6: The near-overlap of blue and red curves makes them indistinguishable from the purple one. I suggest swapping orange and red. This also makes the order of the colours more intuitive.

Thanks, we have swapped the order here and in the new Figure 9 accordingly.

O.302 Is the shallower depth of neutral buoyancy determined by a fresher plume, a warmer plume, or significantly impacted by both?

Because salinity has a much greater influence on density than temperature (for the ranges found in Greenland's fjords), the shallower depth of neutral buoyancy is primarily determined by a fresher plume – this has been added into the explanation here in L.331-333.

O.311-317 It's a bit difficult to follow the reasoning, while the point you try to make is quite straightforward. Please rewrite and to more directly get to the point.

This section has been significantly simplified (L.342-345) to more directly say that (a) the basic shape of the temperature profile is the same regardless of the temperature anomaly and (b) the temperature differential between profiles decreases at shallower depths, which reflects the shape of the initial temperature profile.

O.336 profile -> profiles

Changed in L.375.

O.339 This sentence implies that sublinear means a relatively weak sensitivity. This is not per se the case.

In this context we are using sublinear to convey the fact that the functional dependence is less than linear, rather than to imply a relatively weak sensitivity. Thus we have kept the terminology but have reworded to avoid the implication in L.376-377.

O.340 Can you explain why this sensitivity is sublinear (what is the negative feedback / saturation process that decreases the sensitivity?)

The sensitivity of melt rate to subglacial discharge is sublinear because the volume flux entrained by a plume scales sublinearly with subglacial discharge (e.g. Cowton et al., 2016, Fig. 6b). If the volume flux scales sublinearly then so does the velocity of the resulting fjord circulation (Fig. 6b in the current paper), which then also means that the melt rate scales sublinearly. This has now been made clearer in the text in L.376-378.

O.340 'Approximately linear'. Elsewhere, this is referred to as supralinear, which you indeed quantify in the parameterisation. Here, you give the explanation for the positive feedback inducing the supralinearity. Please link these statements and explanations

together explicitly and make sure that what you interpret as supralinear is referred to as supralinear throughout the paper.

Thank you for this comment – we have changed 'approximately linear' to 'supralinear' here for consistency (L.379).

O.344 **a** higher subglacial discharge

Added "a" in L.383.

Fig 7: The caption isn't fully self-explanatory. Please explain that all variables are averaged horizontally over the full domain. And again, I'd suggest changing 'velocity magnitude' to speed (same for Fig 8)

We had specified this back in the caption for Figure 6, but we agree that the result of this is that Figure 7's caption wasn't fully self-explanatory. We have updated the caption for Figures 7, 8 and 9 to specify that the variables are averaged over the whole area of the mélange.

O.361 'as as' -> 'as a'

Changed in L.407 – thanks for catching this.

O.363-367 I think it's more logical to say that T and S are the 'restoring / forcing' temperature and salinity. Because you state earlier that the results are independent of the initial conditions. The fact that you choose to initialise with the restoring conditions is irrelevant.

We agree – we have changed "initial" to "restoring" (L.409) and "forcing" (L.412) from previous O.363/O.366.

O.363 You have a double use for lambda in the paper. I'd keep this here (usual nomenclature for the freezing point parameters) and change lambda for the iceberg area fraction. (Unless that's also a common usage, in that case, ignore this point)

Unfortunately, lambda is also commonly used as the iceberg area fraction (and also more generally in fields of study involving flow around obstacles, see e.g. Hughes (2022), section 2.1). Therefore, we think it is appropriate to use lambda both ways and hope that the subscripts and limited use here are not too confusing.

O.412 **the** deep ocean temperature

Added "the" in L.522.

O.419 'three times larger'. How does this compare to your results? And if they differ, can you explain why? If you don't compare this to your own results, it's not a relevant result to mention in your discussion.

Apologies that we did not make the comparison explicit in the first manuscript. Enderlin et al. (2016) define shallow-drafted (deep-drafted) icebergs are those icebergs with a draft shallower (deeper) than the Polar-Atlantic Water Interface Depth, which ranges between 115m and 176m (their Table S1). Comparing their results to the standard simulation ($Q_0 = 300 \text{ m}^3\text{s}^{-1}$, Figure 6c), we can see that the average melt rate at 50m depth is approximately a third of the average melt rate at 400m depth. This is

comparable to the results from Enderlin et al. (2016). This has been added to the revised manuscript in L.529-533.

Sec 4.3 You explicitly mention the seasonality in observations. How should I as a reader interpret this? Can your results be compared to the seasonally biased observations? Or are your results representative of annual means? Please provide a bit of perspective

At this point we didn't really mean to get into seasonality, we only wished to provide the months of the Enderlin observations for context. The Enderlin observations come from June, July and October so likely cover a range of subglacial discharge fluxes. Thus we feel it is appropriate to compare her results to the full range of our results and this has now been clarified in the manuscript in L.533-534.

O.431-437 These discussion points are again (strongly) dependent on the choice of lambda. Reconsider how these points should be perceived considering the uncertainty in lambda.

Having added additional simulations with lower values of $\lambda$ to new Figure 10, we believe that these discussion points are still valid because the sensitivity to lambda is not so strong that it affects these points.

O.440 this study -> our study (avoid ambiguity)

Changed in L.550.

O.442 'uses': this is unclear. Do they prescribe velocities, in which case the result (maximum at 100m) is trivial? Or do they restore velocities to this half-sinusoid, in which case the velocities are actually resolved? Please be specific

This is a bit tricky – they prescribe boundary velocities that vary sinusoidally with depth (in order to roughly represent the sheared exchange flow in Greenland fjords), but subsequently find that the velocity maximum in the simulation interior is around the depth of the deepest icebergs. Ideally, we don't want to get into this much detail in the paper as the key point is that Hughes (2022) prescribed the boundary velocity whereas we actually simulate the velocity driven by the subglacial discharge. Therefore, we have clarified that Hughes (2022) prescribes velocities but haven't gone further than that. Hopefully the reviewer agrees this is appropriate.

O.467 'The precise values ... might vary'. If you link the values to physical interpretation (see point above), you may be able to provide useful constraints: that, based on physical reasoning, c will generally be < 1, and d >1.

Thank you for this – we have added a sentence explaining this in L.579-580.

O.478 Can you, based on the sensitivity to mélange thickness, and your analysis of down- fjord gradients in melting, provide some insight into how important you think the exact distribution may be?

We have added some discussion on this in L.439-443.

O.473 (Sorry for going on about this the whole time) A third source of uncertainty is the total areal fraction of icebergs. This deserves a mention here.

We agree and have added this in (L.435 and L.454-462).

O.498 Are you sure this is 'the key limitation/uncertainty'? If so, why did you not do a sensitivity test to this? The simplest reply to this point is to change this to 'a key limitation/uncertainty', but perhaps, based on your knowledge and trial and error, you can provide a bit more insight than that on how important this aspect is.

We have changed "the key" to "a key" in L.476, because you're right, we're not sure this is the key limitation. Our phrasing here came from the fact that melt rate parameterisations do have a lot of uncertainty (see for example the recent series of papers on LeConte Glacier in Alaska (e.g. Jackson et al., 2020)), but we chose not to get into this uncertainty because the set of simulations we did was already computationally expensive and we felt the paper is full enough already. We don't feel we can add too much here beyond the quite extensive discussion that is already in that paragraph.

O.510 This is a contradictory sentence. The constants define the sensitivity, so the sensitivity cannot be independent of these constants.

Our point here is that we believe the functional form of the parameterization developed in this study is independent of the precise values of the constants used in the three-equation melt rate parameterization. In greater detail – varying the constants used in the three-equation melt rate parameterisation would not change the exponents on the subglacial discharge and the thermal forcing in the melt rate parameterisation (Eq. 3). We agree that this was not clear and may on first reading appear contradictory, and we have changed the wording appropriately (L.488-490).

O.531 Here, it's again 'approximately linear' to ocean temperature. Make sure your own interpretation is consistent throughout.

Thank you, we have changed this to supralinearly here (L.598) and throughout to be consistent.

O.535 predict -> project

Changed in L.603.

**Reviewer 2**

**Major comments**

I do not expect the authors to run more simulations but I think that they should make it clear in the methods that because they use the same aspect ratio and areal coverage of icebergs in their three thickness simulations, that they inherently include more icebergs in the thin simulations. The authors discuss how the larger number of icebergs in the thin simulations increases the surface area of the icebergs in the fjord and that brings up the meltwater flux even though the average melt rate is reduced. Does the difference in iceberg abundance also influence the magnitude of horizontal velocities in the simulations? It is not clear if the velocities are approximately the same for all thickness simulations because the areal coverage of icebergs is the same or if the abundance of icebergs influences that parameter as well. I recommend that the authors briefly discuss the likely effects of changes in iceberg abundance on their simulations since it is one of their least constrained

parameters and the thickness and abundance of icebergs will both be less for almost all other fjords around Greenland.

Thank you for your comment – we have added an explicit statement in the methods that there are more icebergs in the thin simulations (L.177). With regards to the impact of iceberg abundance on the simulations, we believe that the additional simulations we have run considering lower values of lambda (discussed extensively above) addresses this. The sensitivity to lambda is discussed in section 3.2.4 and the new simulations demonstrate that changing $\lambda$ has a relatively small effect on the mélange melt rate and so the results from the originally submitted paper still hold.

I'd also like the authors to make it more clear that the iceberg geometries do not evolve in this simulation. You are essentially adding fixed iceberg blocks that do not change shape. I don't take issue with that model configuration but it should be made clear in the method and that simplification should be discussed when you describe the potential influence of iceberg shape on the simulations.

Thank you for highlighting this important point. We have added sentences in L.159-162 making this explicit (and also explaining that the change in iceberg geometries due to melting throughout the model run would be smaller than the resolution of the model). We have also added a brief paragraph in the discussion section (L.472-475).

**Minor comments**

O.56-61: I struggled a bit with this paragraph. The sentences summarizing results of the two Hughes references almost feel contradictory. This paragraph also felt a bit out of place because a lot of the preceding text focused on observations and then subsequent paragraphs focus on modeling. This paragraph describes modeling results but it is not all that obvious that you switched from observations to models. I suggest wrapping this text into one of the later paragraphs in the intro and rephrasing the two Hughes-focused sentences.

Thanks, we have removed this paragraph and incorporated the contents into paragraphs later on in the introduction.

O.68-78: Somewhere in this section or elsewhere in the intro, I recommend that you cite Hester et al. (2021; DOI: 10.1103/PhysRevFluids.6.023802) or FitzMaurice et al. (2017; doi:10.1002/2017GL073585), which both describe how melt rates can vary around icebergs due to differences in velocity across the varying faces.

Thanks, we have now cited both of these papers in L.85.

O.107: Why is x_2 open water? Is that a requirement for stability? Figure 1 is fantastic for visualizing the set-up.

Thank you. x_2 is open water to leave open the possibility of analysing the melt rate of the calving front too. In MITgcm, the melt rate is a property of an ice-adjacent ocean cell, and so leaving x_2 as open water allows us to isolate the effect of the calving front from the effect

of the mélange. We appreciate that this is unrealistic and given that we chose not to focus on the calving front at all in the present study, this is perhaps something we could choose not to do in the future.

O.177: You describe the three-equation parameterization of melt as one that is "a number of constants whose values are plagued with uncertainty". That set up the reader to expect to learn about multiple constants but you only focus on $C_d$. I recommend that you at least list the other constants and their values even if you focus on the one that you consider to be the most important.

We have added a sentence introducing the turbulent transfer coefficents as constants in the three-equation parameterization in L.222-223.

O.137: I also recommend citing Astrom et al. (2021; https://doi.org/10.1017/jog.2021.14) when describing size distributions.

Thank you, we have done this (L.138-139).

Figure 2: This is terrific.

Thank you.

O.223: I don't follow why you list all these subglacial discharge values when you later state that you only use three.

We use all of these subglacial discharge values for the T0, standard thickness simulation, but only use 10, 300, 1000 for different temperatures and thicknesses (e.g. Table C1). We have made this clearer in the text by rewriting section 2.6.

O.251: The comments about the icebergs being a heat sink for the plume make sense but without a simulation of the plume without icebergs, be careful with how you phrase your inference that the icebergs limit the extent of the plume.

Thank you for this, we have rephrased the explanation in L.269-270 accordingly.

O.257: Remove "of red" since that is really specific to the figure description and removing the color makes it more general.

"Of red" deleted.

O.342-346: I had a difficult time following this description. Please try to revise.

Point taken. We have revised this in L.381-387 to make it clear why higher subglacial discharge flux makes the melt rate of the mélange less sensitive to mélange thickness.

O.393: Isn't the parameterized melt rate for the standard thickness lower than the modeled melt rate for both the thicker and thinner simulations? I interpreted the text here as the opposite of what I saw in Figure 10.

Apologies, there was an error in the figure – thanks for spotting it. In the revised figure, the parameterisation overestimates the melt rate for the thin simulations and underestimates the melt rate for the thick simulations.

O.418-424: I think the difference in mélange areas is moreso because Enderlin et al. (2016) did not differentiate icebergs and sea ice, so all the areas in your simulations that are ice-free would be occupied by ice.

We've been back to Enderlin et al. (2016) and in our best judgement they only consider icebergs in the mélange when they calculate their submerged mélange area, and so we feel this point still stands.

---

## Referee Report (RR1)

Second review of 'Modelling ocean melt of ice mélange at Greenland's marine-terminating glaciers'

Jain et al.

Date: 25 June 2025
Assessment: Minor revisions
Reviewed by: Erwin Lambert

I thank the authors for their extensive revision addressing all my remarks appropriately. In particular, I appreciate the additional sensitivity runs to the iceberg fraction which must have been quite a bit of work. To my taste, this revised manuscript conveys the author's results more clearly and convincingly.

Altogether, I consider this manuscript to be a high-quality contribution to *The Cryosphere* and recommend acceptance after minor revisions. Note that the remaining comments I have are very minor and should be easy to address.

Minor comments

Throughout the paper, the ambient or restoring temperature is referred to as initial temperature. As in my previous review: this appears to be inconsistent with your statement that initial conditions are irrelevant (l.130-134). I suggest the authors ctrl+F on 'initial' starting at this statement, and reconsider all following mentions whether this is the most appropriate term. Particularly Sec. 3.2.2 is titled 'ambient temperature', while the authors still refer to this as the initial temperature.

l. 368 The velocities appear to show a minimum for lambda = 0.4, with a peak at both 0.2 and 0.6. My impression is that a low iceberg density facilitates a shallow flow between the icebergs, and a high density facilitates a relatively continuous flow along the bottom of the icebergs. I suggest the authors revise this statement.

---

## Author Response (AR2)

**Introduction**

We thank the reviewers once again for their time and helpful comments. We have addressed all of their minor comments and we respond to all of them below. Reviewer comments are in black; our comments are in blue. References to line numbers in the last iteration of the manuscript are styled L.X; references to line numbers in the tracked-changes version of the last iteration of the manuscript are styled T.X; and references to line numbers in the revised manuscript are styled R.X.

**Reviewer 1**

**Minor comments**

Throughout the paper, the ambient or restoring temperature is referred to as initial temperature. As in my previous review: this appears to be inconsistent with your statement that initial conditions are irrelevant (L.130-134). I suggest the authors ctrl+F on 'initial' starting at this statement, and reconsider all following mentions whether this is the most appropriate term. Particularly Sec. 3.2.2. is titled 'ambient temperature', while the authors still refer to this as the initial temperature.

We agree with the reviewer that referring to the temperature profile as the "initial" temperature is inconsistent with our statement that the initial conditions are irrelevant. We have therefore taken their advice and replaced all instances of "initial" with either "ambient" (Figure 1 caption, R.292, R.323, R.342, R.347) or "restoring temperature profile" (R.270, R.349). The only section in which we have not done this is section 2.2 "Initial and boundary conditions", since it is this section in which we explain that the initial conditions are irrelevant.

L.368 The velocities appear to show a minimum for lambda = 0.4, with a peak at both 0.2 and 0.6. My impression is that a low iceberg density facilitates a shallow flow between the icebergs, and a high density facilitates a relatively continuous flow along the bottom of the icebergs. I suggest the authors revise this statement.

We agree with the reviewer's interpretation and have adjusted the text accordingly in R.366-369. We have also replaced instances of "velocity" with "flow speed" to be consistent with Figures 6-9 and in light of the reviewer's comment on the last iteration of the manuscript.

**Reviewer 2**

**Minor comments**

There are a few places where the new text could use some revision, as I point out below. Line numbers are from the document with tracked changes.

We thank the reviewer for taking the time to suggest alternative phrasing for sections of the text. We have adopted all of their suggestions, as detailed below.

T.13-14: "The magnitude of the simulated melange melt rate ranges from 0.26 m d^-1 to 0.92 m d^-1, which is in good agreement with observational estimates."

We have edited this as requested in R.13-14.

T.88-91: I recommend that you replace these lines with the following or something similar that more seamlessly incorporates the new references: "The melt rate of ice depends on the water velocity (Holland and Jenkins, 1999), such that variations in the magnitude and direction of water with respect to iceberg faces can influence their rate of melt (FitzMaurice et al., 2017; Hester et al., 2021). Therefore, it is likely that the distribution of icebergs within melange influences water velocity and iceberg melt rate."

We have edited this as requested in R.82-85.

T.167: Remove the sentence that starts with "Note that the iceberg…" and edit the following sentence to weave in that statement: "In light of each simulation run time (section 2.6) and the modelled melt rates (section 3.1.2), changes in iceberg geometry over time are less than the model resolution and are ignored in our simulations."

We have edited this as requested in R.159-160, with the small addition of "…and are **thus** ignored in our simulations."

T.187-191: I noted the inclusion of growlers, bergy bits, and sea ice in the melange area estimates of Enderlin et al. (2016) in my last review but my comment was anonymous so the authors relied on their interpretation of my study instead of my comment. Helheim's dense melange is likely mostly composed of small iceberg fragments, but the method that I used to estimate melange area included these fragments and sea ice, meaning there was no open ocean between the bigger icebergs. So, at least a portion of your under-estimate is due to the inclusion of small iceberg fragments and sea ice in Enderlin et al. (2016) and you should note it here because it means your big iceberg estimates are probably more similar in area to my observation-based estimates that you give credit for.

We thank the reviewer for providing more detail and insight into the methodology behind their study. We have added the point that the inclusion of small iceberg fragments by Enderlin et al. (2016) in their calculation of submerged mélange area likely contributes to the fact that the value of our simulated area is lower. We have included this point in R.177-178.

T.328-334: The second sentence in this paragraph is VERY long and complex. I'd break it apart. Also, you say "this along-fjord variation is averaged over" but it is not clear what it is averaged over. It should say it is averaged over a specific length scale.

We accept the point that this sentence was very long and therefore unclear. We have broken up and edited it slightly in R.311-314. We have also specified that the along-fjord variation is averaged over the full length of the mélange.

T.406-407: Revise and condense the last two paragraphs of this sentence to be something like "As a result, thermal forcing is relatively uniform with depth and melt rates are similar for shallow- and deep-drafted icebergs for high values of subglacial discharge."

We have edited this as requested in R.383-385.